# Dual neutrophil subsets exacerbate or suppress inflammation in tuberculosis via IL-1β or PD-L1

Emilie Doz-Deblauwe[1],*, Badreddine Bounab[1],*, Florence Carreras[1], Julia S Fahel[1,5], Sergio C Oliveira[4,5], Mohamed Lamkanfi[6], Yves Le Vern[1], Pierre Germon[1], Julien Pichon[1], Florent Kempf[1], Christophe Paget[2,3], Aude Remot[1], Nathalie Winter[1]

**Neutrophils can be beneficial or deleterious during tuberculosis (TB). Based on the expression of MHC-II and programmed death ligand 1 (PD-L1), we distinguished two functionally and transcriptionally distinct neutrophil subsets in the lungs of mice infected with mycobacteria. Inflammatory [MHC-II⁻, PD-L1$^{lo}$] neutrophils produced inflammasome-dependent IL-1β in the lungs in response to virulent mycobacteria and "accelerated" deleterious inflammation, which was highly exacerbated in IFN-γR$^{-/-}$ mice. Regulatory [MHC-II⁺, PD-L1$^{hi}$] neutrophils "brake" inflammation by suppressing T-cell proliferation and IFN-γ production. Such beneficial regulation, which depends on PD-L1, is controlled by IFN-γR signaling in neutrophils. The hypervirulent HN878 strain from the Beijing genotype curbed PD-L1 expression by regulatory neutrophils, abolishing the braking function and driving deleterious hyperinflammation in the lungs. These findings add a layer of complexity to the roles played by neutrophils in TB and may explain the reactivation of this disease observed in cancer patients treated with anti-PD-L1.**

## Introduction

Tuberculosis (TB) is among the principal causes of death because of infectious diseases in the world. This situation was worsened by the recent burden imposed on healthcare systems by the COVID-19 crisis, which severely affected TB management programs ([1]). Almost all human cases of TB are due to *Mycobacterium tuberculosis* (Mtb). The first laboratory strain sequenced in 1998 by Cole et al was H37Rv ([2]). It was long believed that genetic diversity among Mtb strains was limited. The recent development of whole-genome sequencing uncovered the complex geographical distribution of nine different phylogenetic lineages (L) of Mtb circulating in different regions of the world ([3]). The L2 and L4 strains are the most highly distributed

worldwide, with the L2 strain dominating in East Asia, with high transmission rates. Most experimental TB pathophysiology studies have been conducted using the laboratory-adapted L4 strain H37Rv. However, strains from different lineages induce different pathological spectra in humans and animal models ([4]). HN878, the prototype L2 "Beijing" hypervirulent strain, causes an exacerbated immunopathology. However, the immune mechanisms underlying such severe disease are not fully understood.

After infection with Mtb, most people do not develop immediate signs of disease but may remain latently infected for decades. During this period, a status quo between the host and the bacilli involves several immune mechanisms to regulate host defense and inflammation. The role of the programmed death 1/programmed death ligand 1 (PD-1/PD-L1) axis in restricting T-cell function has been recently highlighted. Blockade of these immune checkpoints has brought considerable progress to cancer treatment in recent years ([5]). However, concerns are now emerging about an increase in active TB cases after such treatment ([5], [6]). Experimentally Mtb-infected PD-1–deficient mice quickly die ([7]) because of the detrimental overproduction of pathogenic IFN-γ by CD4⁺ T cells in the lung parenchyma ([8]). Mtb-infected rhesus macaques treated with anti-PD-1 develop exacerbated disease, which is linked to caspase-1 activation ([9]). Inherited PD-1 deficiency in humans is linked to decreased self-tolerance and anti-mycobacterial immunity ([10]).

The hallmark of TB is the formation of granulomas in the lung; in these organized pluricellular structures, a delicate balance between the containment of Mtb replication and host inflammation takes place. The fate of Mtb, from eradication to active multiplication, may vary depending on the granuloma microenvironment, where multiple immune mechanisms are at play to maintain or disrupt immunoregulation ([11]). Among innate cells, neutrophils play dual roles in TB ([12]). At early stages, they halt Mtb infection and shape early formation of the TB granuloma ([13], [14]). At later stages, their highly destructive arsenal is critical for TB reactivation; they represent the first expectorated cells of active TB patients ([15]). In the

[1]INRAE, Université de Tours, Nouzilly, France    [2]INSERM, U1100, Centre d'Étude des Pathologies Respiratoires, Tours, France    [3]Faculté de Médecine, Université de Tours, Tours, France    [4]Department of Immunology, University of Sao Paolo, Sao Paulo, Brazil    [5]Department of Biochemistry and Immunology, Federal University of Minas Gerais, Belo Horizonte, Brazil    [6]Laboratory of Medical Immunology, Department of Internal Medicine and Pediatrics, Ghent University, Ghent, Belgium

Correspondence: nathalie.winter@inrae.fr
*Emilie Doz-Deblauwe and Badreddine Bounab contributed equally to this work

mouse, we have shown that neutrophils reach the lungs in two waves during the establishment of the immune response, with the adaptive wave playing no role in Mtb growth restriction [16]. There is now extensive evidence that neutrophils represent a heterogeneous and plastic cellular compartment [17]. Some neutrophils are endowed with classical phagocytic and pathogen-killing functions, whereas others are able to cross-talk with a variety of immune cells, taking full part in the adaptive immune response [12]. In this context, we have recently characterized a new subset of regulatory neutrophils that can be functionally distinguished from classic neutrophils in healthy cattle and mice by their ability to suppress T-cell proliferation [18]. Despite the recognized role of neutrophils in TB, the potential role of this new subset has not been explored yet.

IL-1$\beta$ is a cornerstone cytokine in TB. It is essential for constraining Mtb infection in the early stages, as unequivocally demonstrated in mouse models, but may also become deleterious at later stages of the full-blown adaptive immune response. Cross-regulatory pathways of IL-1$\beta$ production during TB include that of type I IFN, which directly down-regulates pro-IL-1$\beta$ gene transcription [19]. Bioactive IL-1$\beta$ needs to be processed from immature pro-IL-1$\beta$ via inflammasome assembly, to which macrophages (MPs) are the major contributor. In vitro, in response to Mtb infection, bone marrow–derived MPs assemble the NLRP3 inflammasome and activate caspase-1 to trigger canonical inflammasome activation and the release of mature IL-1$\beta$ [20]. Beyond MPs, recent studies suggest a role of NLRP3 inflammasome–dependent IL-1$\beta$ production by neutrophils in vivo [21]. However, the contribution of neutrophils to IL-1$\beta$ production during TB appears to be much less than that of MPs [22] and it is assumed that caspase-1–independent mechanisms account for pro-IL-1$\beta$ cleavage by these cells [23].

As neutrophils shape the fate and full development of granulomas during TB disease, we revisited the role of these heterogeneous plastic cells [17, 18] during mycobacterial infection. We compared the recruitment and functions of neutrophil subsets during infection with the avirulent live vaccine BCG and the two virulent Mtb strains, H37Rv (L4, laboratory-adapted) and HN878 (L2, Beijing prototype). We used the IFN-$\gamma$R$^{-/-}$ mouse model, in which extensive neutrophil-driven inflammation was described [24] before distinct subsets were known. We also analyzed the potential for inflammasome-dependent mature IL-1$\beta$ production by neutrophils in vitro, as well as in vivo, taking advantage of a new mouse model in which caspase-1–dependent IL-1$\beta$ secretion is specifically abrogated in neutrophils [25]. We provide evidence that distinct subsets play opposite roles in TB pathophysiology by contributing to IL-1$\beta$–driven inflammation in the lungs or regulating neutrophilia via the immune checkpoint inhibitor PD-L1.

# Results

## The neutrophil NLRP3 inflammasome contributes to IL-1$\beta$ production during mycobacterial infection

We infected neutrophils from mouse bone marrow with the avirulent vaccine strain BCG and the virulent H37Rv and HN878 Mtb strains. Different MOIs for the BCG (10:1) virulent Mtb (1:1) were used to preserve neutrophil viability. This induced a comparable release of TNF (Fig S1A). Neutrophils also released mature IL-1$\beta$ in response to infection by all strains (Fig 1A), albeit to a lesser extent than after LPS plus nigericin stimulation. We next prepared neutrophils from the bone marrow of various genetically deficient mice to test the role of the inflammasome. IL-1$\beta$ secretion by mycobacteria-infected or LPS/nigericin-stimulated neutrophils from Nlrp3$^{-/-}$ (Fig 1B), Csp1/11$^{-/-}$ (Fig 1B), and Gsdmd$^{-/-}$ (Fig 1C) mice was severely impaired relative to that of WT neutrophils. This was not due to activation issues, as these genetically deficient neutrophils released similar levels of TNF (Fig S1B). Canonical assembly of the inflammasome and pyroptosis appears to be involved in the IL-1$\beta$ maturation process in neutrophils. This was confirmed with neutrophils from Csp11$^{-/-}$ mice, which secreted similar levels of mature IL-1$\beta$ as WT mice in response to BCG infection (Fig S1C). Of note, neutrophils from Aim2$^{-/-}$ or WT mice produced similar levels of mature IL-1$\beta$ (Fig 1C). We observed the cleavage of pro-IL-1$\beta$ into mature IL-1$\beta$ of 17 kD in neutrophils infected with BCG (MOI 20:1) by Western blotting (Fig 1D), confirming inflammasome assembly. MPs secreted more mature IL-1$\beta$ into the supernatant than neutrophils (Fig 1E), regardless of the stimulus. On a cell-to-cell basis, MPs secreted 47 times more mature IL-1$\beta$ than neutrophils after infection with virulent Mtb H37Rv and 35 times more than that after BCG stimulation (Fig 1E).

We assessed the contribution of neutrophils to IL-1$\beta$ production in vivo by infecting mice with virulent Mtb H37Rv and injecting the anti-Ly-6G antibody at the onset of recruitment of the second wave of neutrophils, that is, between days 17 and 21 [16]. This treatment markedly reduced the number of neutrophils in the lungs (Figs 2A and S2A for gating strategy). Lesions were more extensive in anti-Ly-6G than isotype-treated mice (Fig 2B), with a twofold greater total lung surface occupied by the lesions (Fig 2C). Production of IL-1$\beta$ in the lung tissue of anti-Ly-6G–treated mice was 2.3-fold less than that in the lung tissue of mice injected with the isotype control antibody (Fig 2D). These data thus confirm the role of neutrophils in the formation of lung lesions during Mtb infection [12] and indicate their direct participation in IL-1$\beta$ production in vivo. We confirmed this using the recently obtained MRP8$^{Cre+}$Csp1$^{flox}$ mouse strain [25], in which IL-1$\beta$ production is specifically abolished in neutrophils. We first validated this tool in vitro using purified neutrophils and bone marrow–derived MPs from MRP8$^{Cre+}$Csp$^{flox}$ mice and their MRP8$^{WT}$Csp1$^{flox}$ littermates stimulated with LPS and nigericin. As expected, neutrophils from MRP8$^{Cre+}$Csp1$^{flox}$ mice did not produce IL-1$\beta$, whereas MRP8$^{WT}$Csp1$^{flox}$ neutrophils did (Fig 2E). In addition, MPs from MRP8$^{Cre+}$Csp1$^{flox}$ and MRP8$^{WT}$Csp1$^{flox}$ mice equally produced IL-1$\beta$, as expected (Fig 2F). Next, we intranasally infected MRP8$^{Cre+}$Csp1$^{flox}$ mice and their MRP8$^{WT}$Csp1$^{flox}$ littermates with Mtb H37Rv and observed significantly lower IL-1$\beta$ production in the lungs of the MRP8$^{Cre+}$Csp1$^{flox}$ than those of the MRP8$^{WT}$Csp1$^{flox}$ animals (Fig 2G). This result confirms the direct participation of neutrophils in IL-1$\beta$ production after inflammasome assembly in the lungs in response to Mtb infection.

## Mycobacteria attract inflammatory and regulatory neutrophil subsets to the lung

We recently discovered that [Ly-6G$^+$, MHC-II$^-$, PD-L1$^{lo}$] neutrophils, akin to classic neutrophils, and [Ly-6G$^+$, MHC-II$^+$, PD-L1$^{hi}$] regulatory

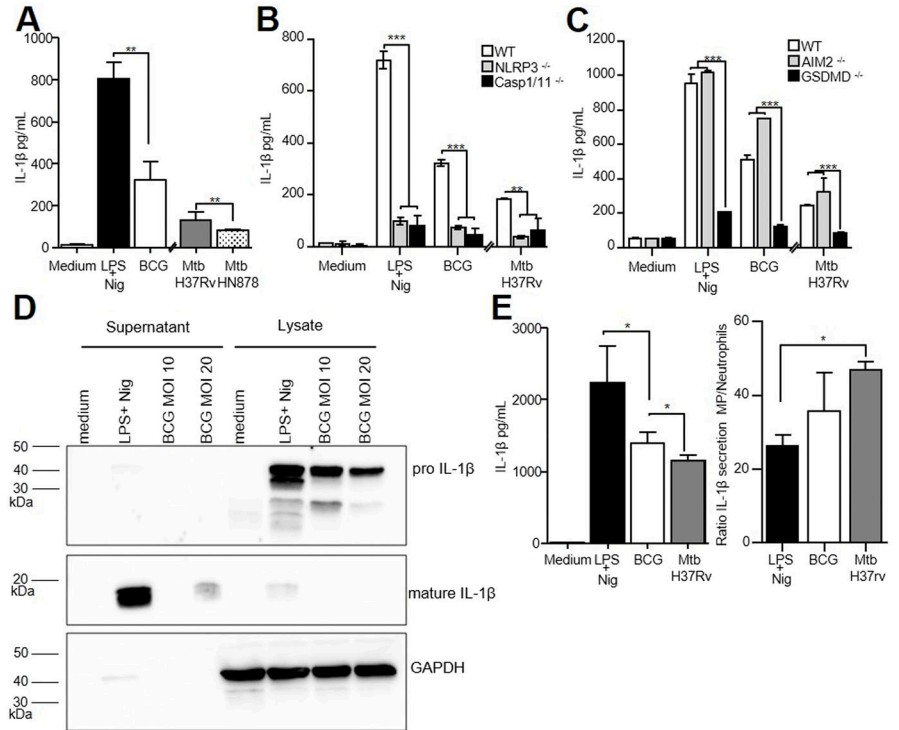

**Figure 1. Neutrophil NLRP3 inflammasome contributes to IL-1β production after mycobacterial infection.**
**(A, B, C)** Mature IL-1β produced by WT (A), *Nlrp3*[−/−], *Csp1/11*[−/−] (B), *Aim2*[−/−], and *Gsdmd*[−/−] (C) bone marrow neutrophils was determined by ELISA after overnight stimulation with LPS/nigericin or infection with BCG (MOI 10) or Mtb (H37Rv or HN878, MOI 1). **(D)** Immunoblotting of pro-IL-1β, mature IL-1β, and GAPDH in supernatants and lysates from bone marrow neutrophils infected for 5 h with BCG (MOI 10 or 20) or stimulated with LPS/nigericin. **(E)** Mature IL-1β produced by WT bone marrow–derived MPs was determined by ELISA after overnight stimulation with LPS/nigericin or infection with BCG (MOI 10) or Mtb H37Rv (MOI 1). **(A, B, C, D, E)** Pooled data from three independent experiments, n = 6 mice; (B) data are representative of three independent experiments, n = 3; (C) data are representative of two independent experiments, n = 3; (D) data are representative of two independent experiments, n = pool of 10 mice; (E) pooled data from two independent experiments, n = 4. **(A, B, C, E)** Graphs show medians with ranges. ∗P < 0.05, ∗∗P < 0.01, and ∗∗∗P < 0.001 by the Mann–Whitney test (A, E) and the non-parametric Fisher–Pitman permutation test (B, C).

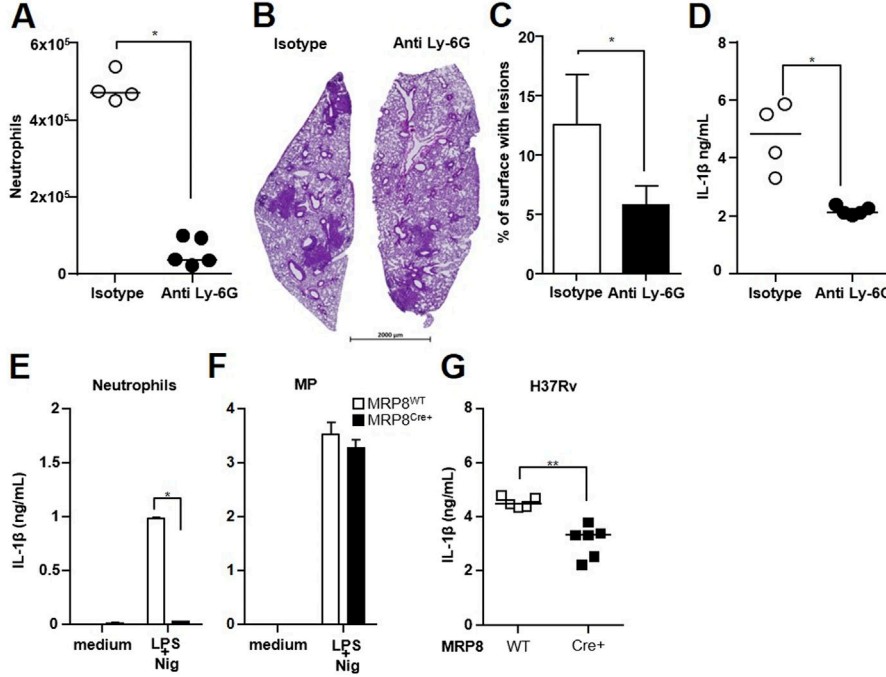

**Figure 2. Neutrophils directly contribute to IL-1β production in the lungs during H37Rv infection.**
**(A, B, C)** C57BL/6 mice were infected with 10[3] CFUs of Mtb H37Rv and neutrophils depleted by intraperitoneal administration of anti-Ly-6G or isotype control antibody on days 15, 17, and 19. Lungs were harvested on day 21 for analysis. **(A)** Total lung neutrophils were identified by flow cytometry as CD11b[+] Ly-6G[+] Ly-6C[+] cells (see Fig S2A for gating strategy). **(B)** Representative image of hematoxylin/eosin staining of lung sections. **(C)** Total lung surface occupied by lesions in lung sections from anti-Ly-6G or isotype control antibody-treated mice. **(D, E, F)** IL-1β production in the lung tissue from C57BL/6 mice or in the supernatants after overnight stimulation with LPS/nigericin of bone marrow neutrophils (E) or MPs (F) from *MRP8*[Cre+]*Csp1*[flox] or *MRP8*[WT]*Csp1*[flox] mice measured by ELISA. **(G)** IL-1β production in lung tissue from *MRP8*[Cre+]*Csp1*[flox] or *MRP8*[WT]*Csp1*[flox] mice 21 d post-infection with H37Rv measured by ELISA. **(A, B, C, D, E, F, G)** One experiment, n = 4–5 mice per group; (E, F) two independent experiments, n = 4 mice; (G) data are representative of two independent experiments, n = 5–6 per group. Medians with ranges (C, E, F) and individual data points with medians for (A, D, G). ∗P < 0.05 and ∗∗P < 0.01 by the Mann–Whitney test (A, B, C, D) and the non-parametric Fisher–Pitman permutation test (E, F).

neutrophils circulate in blood as two functionally different subsets at the steady state in healthy mice and cattle. Only regulatory neutrophils are able to suppress T-cell proliferation (18). Thus, we

first assessed the recruitment of these two neutrophil subsets to the lungs after intranasal infection with 5 × 10[6] CFUs of avirulent BCG. Surface MHC-II was used to discriminate between classic and

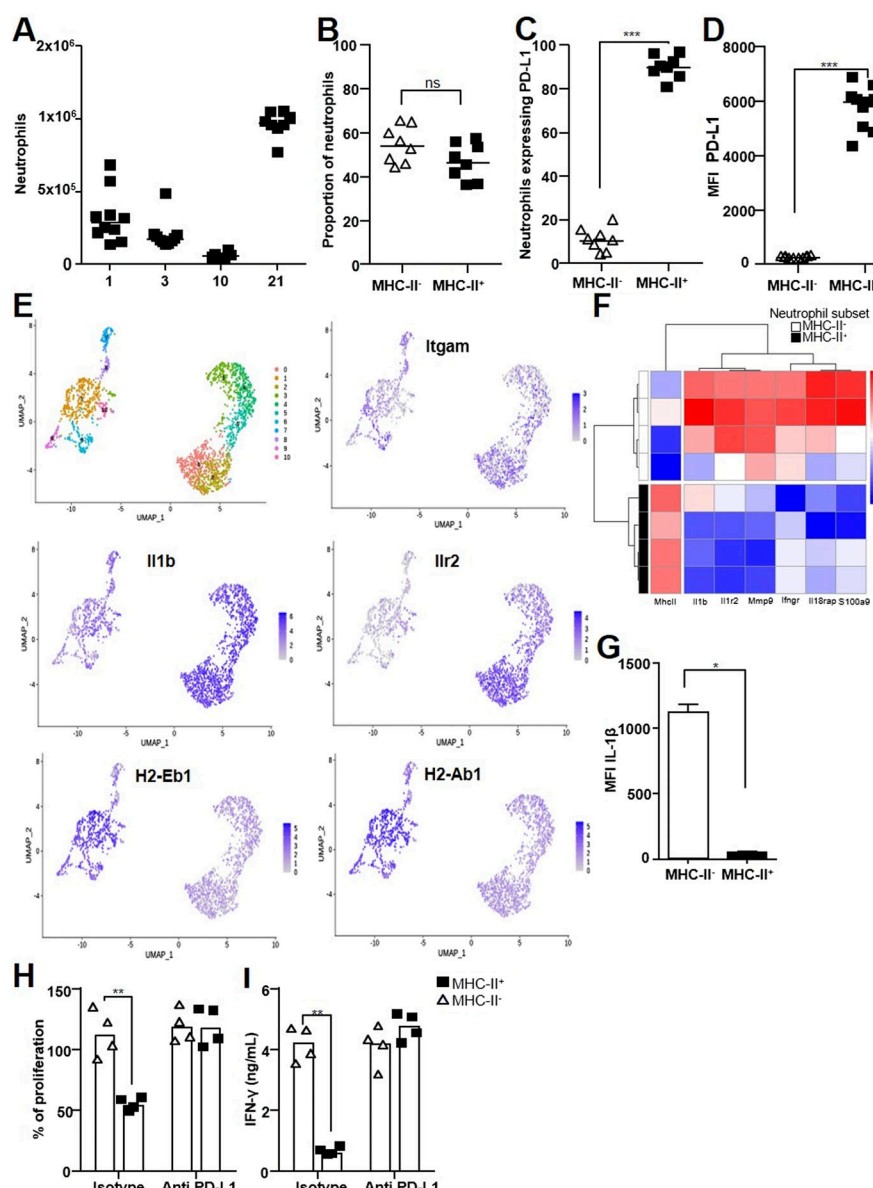

**Figure 3. Mycobacteria attract inflammatory and regulatory neutrophil subsets to the lung.**
**(A, B, C, D, E)** C57BL/6 WT mice were infected with BCG and the lungs processed for the following analysis. **(A)** Kinetics of total lung neutrophils recruited to the lungs at days 1, 3, 10, and 21 post-infection assessed by flow cytometry (Fig S2A for gating strategy). **(B, C, D, E, F)** Lung neutrophils were further characterized on day 21 post-infection. **(B)** Proportion of MHC-II+ and MHC-II− neutrophils among total lung neutrophils. **(C)** Percentage of PD-L1 expression among neutrophils in each subset. **(D)** PD-L1 mean fluorescence intensity in each subset. **(E)** Single-cell RNAseq analysis of Ly-6G+ neutrophils purified from a pool of lung cells from 10 mice. Identification of 11 cell clusters using the SEURAT package on Uniformed Manifold Approximation and Projections. Each dot represents one cell. Visualization of *Itgam*, *Il-1b*, *Il-1R2*, *H2-Eb1*, and *H2-Ab1* gene expression in the clusters as analyzed by the SEURAT package. **(F)** Heatmap representation of differential gene expression between MHC-II− and MHC-II+ neutrophils. **(G)** C57BL/6 WT mice were infected with Mtb H37Rv, and the mean fluorescence intensity of intracellular IL-1β was measured by flow cytometry in both MHC-II− and MHC-II+ lung neutrophil subsets on day 21. **(H)** MHC-II− and MHC-II+ neutrophil subsets were enriched by magnetic beads from the lungs of H37Rv-infected mice on day 21 and mixed with OT-II cells. The percentage of OT-II splenocyte proliferation in the presence of each neutrophil subset was calculated based on proliferation with the Ova peptide only. Neutrophils were treated for 1 h before incubation with anti-PD-L1 Ab (atezolizumab) or an isotype control. **(I)** IFN-γ production in supernatants of OT-II splenocytes measured by ELISA. **(A, B, C, D, E, F, G, H, I)** Pooled data from two independent experiments (n = 8–10 per group); (B, C, D) pooled data from two independent experiments (n = 8 per group); (E) one experiment, pool of 10 mice; (F) four cell sorting experiments were performed from a pool of five infected animals each time; (G, H, I) pooled data from two independent experiments (n = 4). Data are presented as individual data points and medians. ∗*P* < 0.05, ∗∗*P* < 0.01, and ∗∗∗*P* < 0.001 by the Mann–Whitney test (A, B, C, D, G) and the non-parametric Fisher–Pitman permutation test (H, I).

regulatory neutrophils by flow cytometry (Fig S2A). As previously observed (16), total [CD45+, CD11bhi; Ly-6Ghi, Ly-6C+] neutrophils peaked in the lungs 21 d after BCG infection (Fig 3A), together with T cells. This cell population was composed of a balanced mix of [Ly-6G+, MHC-II−] classic neutrophils and [Ly-6G+, MHC-II+] regulatory neutrophils (Fig 3B), which showed similar morphology (Fig S2B). PD-L1 also clearly distinguished [Ly-6G+, MHC-II−, PD-L1lo] from [Ly-6G+, MHC-II+, PD-L1hi] neutrophils (Fig S2A). Overall, 90% of [Ly-6G+, MHC-II+] regulatory neutrophils were PD-L1hi and 10% of [Ly-6G+, MHC-II−] classic neutrophils were PD-L1lo (Fig 3C). Moreover, the MFI of [Ly-6G+, MHC-II+, PD-L1hi] neutrophils was 26 times higher than that of [Ly-6G+, MHC-II−, PD-L1lo] neutrophils (Fig 3D). We then performed single-cell RNAseq analysis of the total Ly-6G+ neutrophil population purified from the lungs 21 d after BCG infection and observed distinct transcriptional profiles (Fig 3E). SEURAT

software classified RNA expression into clusters numbered 0 to 10 (Fig 3E, first panel) that formed two main groups: clusters 0, 2, 3, 4, and 5 formed part of one pool (Fig 3E, right pool), whereas clusters 1, 6, 7, 8, 9, and 10 formed another (Fig 3E, left pool). Of note, SingleR software, trained on the Immunologic Genome Project database of mRNA profiles, identified cells from the right pool as "neutrophils," whereas cells from the left pool were identified as "monocytes/ macrophages," probably because of the expression of genes such as *Mhc-II* and *Cd274* (encoding PD-L1). Certain genes, such as *Itgam* (Fig 3E), were similarly expressed in clusters from the two pools, in agreement with the neutrophil signature. However, the differential expression of the *H2-Eb1* and *H2-Ab1* genes from the MHC-II complex or *Il1b* and *Il1r2* inflammatory genes clearly segregated between the two pools (Fig 3E). We confirmed the differential gene expression between classic and regulatory neutrophils by

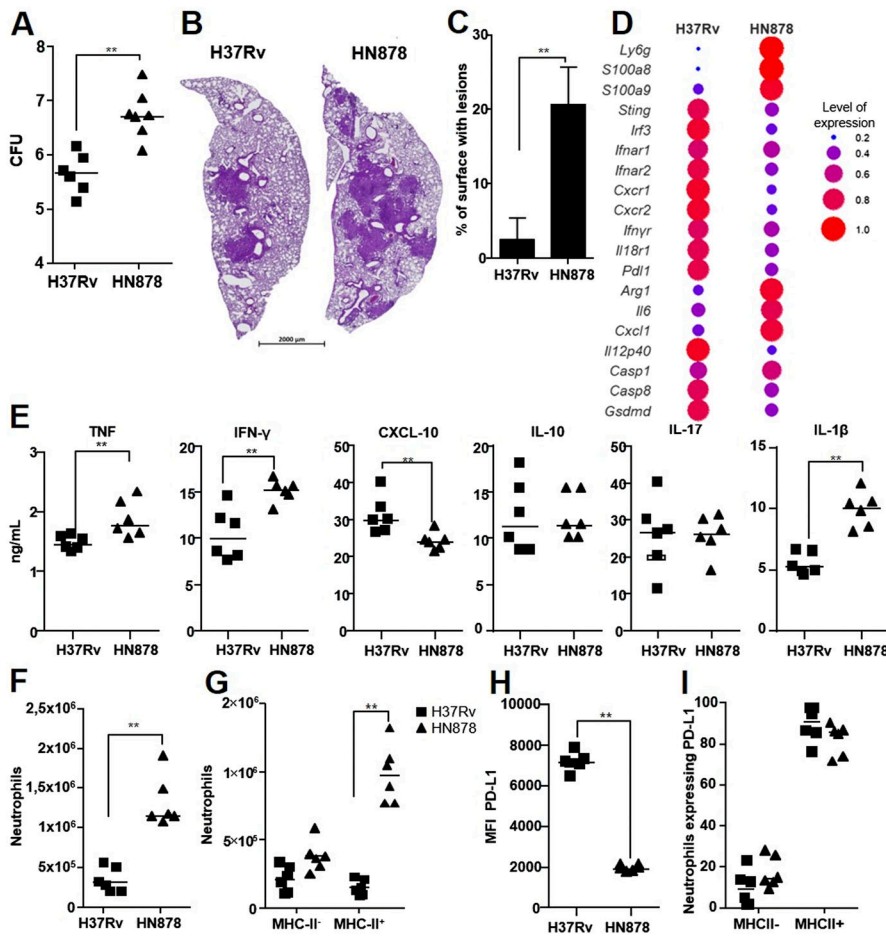

**Figure 4. Two neutrophil subsets are modulated by M. tuberculosis virulence.**
All data (n = 6 mice per group, two independent experiments) were obtained for the lungs of C57BL/6 WT mice at 21 d post-infection with Mtb H37Rv or HN878. **(A)** Number of Mtb CFUs in the lungs. **(B, C)** Representative image of hematoxylin/eosin staining of lung sections for each infected group and (C) the mean percentage of lung surface occupied by lesions. **(D)** Differential gene expression between the two infected groups of a panel of 48 genes normalized against uninfected controls. The dot plot represents the normalized expression of significantly deregulated genes, expressed as the normalized rate to compare the two groups. Data are presented as the mean of n = 4 mice per group from one experiment. **(E)** Cytokine production as analyzed by ELISA in lung tissue homogenates. **(F, G, H, I)** Total neutrophils, (G) neutrophil subsets, (H) PD-L1 surface expression by MHC-II⁺ neutrophils, and (I) percentage of PD-L1^pos among MHC⁺ and MHC-II⁻ neutrophils from the two groups measured by flow cytometry. **(B)** Data are presented as individual data points and medians ((B), median with range). **(A, B, C, D, E, F, G, H)** Two independent experiments, n = 6 mice per group. *P < 0.05, **P < 0.01, and ***P < 0.001 by the Mann–Whitney test (A, B, C, D, E, F, H) and the non-parametric Fisher–Pitman permutation test (G).

performing qRT–PCR targeting genes involved in general neutrophil-driven inflammation, as well as *pro-IL-1β* synthesis and inflammasome assembly. *Mhc-II* genes were expressed in regulatory neutrophils only (Fig 3F). We observed full differential clustering of the two subsets based on the expression of *Il1b, Il1r2, Mmp9, Ifngr, Il18rap,* and *S100a9*; the expression of all these genes was higher in [Ly-6G⁺, MHC-II⁻, PD-L1^lo] classic neutrophils than [Ly-6G⁺, MHC-II⁺, PD-L1^hi] regulatory neutrophils (Fig 3F). We separated the two subsets from H37Rv-infected mice using magnetic beads and observed higher *Il1b* and *Ilr2* gene expression by classic than regulatory neutrophils (Fig S2C). We then analyzed intracellular mature IL-1β production in the lungs ex vivo by flow cytometry 3 wk after H37Rv infection (Fig S2D). The MFI for IL-1β was 25 times higher in [Ly-6G⁺, MHC-II⁻] classic than [Ly-6G⁺, MHC-II⁺] regulatory neutrophils (Fig 3G). Based on these data, we considered classic neutrophils to be "inflammatory" during mycobacterial infection, as documented by both their transcriptional profile and their ability to produce mature IL-1β in vivo.

As the immune inhibitory checkpoint PD-L1 is involved in T-cell suppression (26), we assessed its role in lung regulatory neutrophils recruited in response to mycobacterial infection. We separated regulatory from inflammatory neutrophils from the lungs of BCG- or Mtb H37Rv-infected mice on day 21 and tested their suppressive function ex vivo on splenocytes from OT-II mice ([Fig S2E] and [18]). Only the [Ly-6G⁺, MHC-II⁺, PD-L1^hi] regulatory neutrophils were able to decrease OT-II cell proliferation (by 50%, Fig 3H) and IFN-γ production (by 87%, Fig 3I). We observed similar levels of T-cell suppression by lung regulatory neutrophils obtained from BCG- (Fig S2F) or Mtb H37Rv-infected mice (Fig 3H). Moreover, the addition of the anti-PD-L1 antibody atezolizumab (27) to the wells with regulatory neutrophils obtained from H37Rv (Fig 3H)- or BCG (Fig S2F)-infected mice fully restored proliferation and IFN-γ production by OT-II cells. Thus, only regulatory neutrophils were able to dampen T-cell function and PD-L1 played a major role in this effect.

## The two neutrophil subsets are modulated by *M. tuberculosis* virulence

We next sought information on the role of the neutrophil subsets in TB pathophysiology by comparing lung infection by H37Rv and HN878 on day 21. The number of bacilli in the lungs was 1.1 log₁₀ higher (Fig 4A), and the lesions (Fig 4B) occupied 4.7 times more lung surface (Fig 4C) in HN878- than H37Rv-infected animals, in agreement with the hypervirulence of the Beijing strains (28, 29). However, all mice were clinically stable until the end of our study, that is,

day 21 (data not shown). An analysis of differentially expressed genes (Fig 4D) showed the higher expression of *Sting1, Irf3, Ifnar1,* and *Ifnar2* from the type I IFN pathway in H37Rv- than HN878-infected animals. On the contrary, the expression of the neutrophil marker genes *S100a8* and *S100a9* was higher in the lungs of HN878- than those of H37Rv-infected mice. However, the genes involved in inflammasome assembly and the IL-1β production pathway were not distinctly induced by the two virulent Mtb strains. At the protein level, TNF, IFN-γ, and IL-1β levels were higher in the lungs of HN878 than those of H37Rv-infected mice (Fig 4E), in agreement with the strong inflammatory profile of the strain. On the contrary, levels of CXCL10, a promising biomarker of Mtb infection (30), were lower in HN878- than in H37Rv-infected mice, which could be linked to the type I IFN signature. Hypervirulence of HN878 is linked to strong neutrophilia (31). Indeed, HN878 induced higher recruitment of total neutrophils to the lungs than H37Rv (Fig 4F). Interestingly, the neutrophil influx was composed of 59% inflammatory and 41% regulatory neutrophils after H37Rv infection, whereas HN878 infection resulted in an opposite balance of 71% regulatory and 29% inflammatory neutrophils (Fig 4G). Moreover, infection with HN878 induced a mean MFI for PD-L1 on lung regulatory neutrophils that was 3.8 times lower than that for those of H37Rv-infected animals (Fig 4H) despite a similar frequency of PD-L1 expression among MHC-II[pos] neutrophils between the two strains (Fig 4I). Thus, HN878 was able to downmodulate PD-L1 expression on regulatory neutrophils.

## Caspase-dependent production of IL-1β by inflammatory neutrophils sustains lung inflammation

Inflammatory neutrophils produced mature IL-1β after NLRP3 inflammasome assembly in vivo. We next addressed their contribution to IL-1β–mediated pathophysiology in *MRP8^{Cre+}Csp1^{flox}* mice. We intranasally infected *MRP8^{Cre+}Csp1^{flox}* mice and *MRP8^{WT}Csp1^{flox}* littermates with avirulent BCG and the two virulent H37Rv and HN878 Mtb strains and analyzed their response in the lungs 3 wk later. We first observed that in response to BCG, IL-1β levels in whole lung tissue homogenates were low and comparable in *MRP8^{Cre+}Csp1^{flox}* and *MRP8^{WT}Csp1^{flox}* animals (Fig S3A). On the contrary, IL-1β production by the lungs was lower in *MRP8^{Cre+}Csp1^{flox}* than *MRP8^{WT}Csp1^{flox}* animals in response to the two virulent Mtb strains. Although the response to H37Rv in terms of the amount of IL-1β in the lungs of *MRP8^{Cre+}Csp1^{flox}* mice was only 30% lower than that of *MRP8^{WT}Csp1^{flox}* control mice, it was reduced by 64% in response to HN878 (Fig 5A). Neutrophil-derived IL-1β had no impact on the number of CFUs after infection with BCG (Fig S3B), H37Rv, or HN878 (Fig 5B) at the time point examined. We next examined the differential lung gene expression profile between *MRP8^{Cre+}Csp1^{flox}* and *MRP8^{WT}Csp1^{flox}* mice after infection with BCG (Fig S3C) or the virulent Mtb strains (Fig 5C) on day 21. In response to the three strains, *Cxcl5*, a critical gene for neutrophil recruitment to the lungs (16, 32), and *Cxcl10* were more highly expressed when inflammatory neutrophils were able to produce IL-1β than when they were defective. In addition, HN878 induced higher transcription of *Il-10* and *Cxcr1* when inflammatory neutrophils were defective for IL-1β production.

After BCG instillation, total leukocyte numbers in the lungs were not significantly different between *MRP8^{Cre+}Csp1^{flox}* and

*MRP8^{WT}Csp1^{flox}* mice (Fig S3D), which correlated with no difference in IL-1β production. In response to H37Rv, total lung leukocyte numbers were 24% lower for *MRP8^{Cre+}Csp1^{flox}* than for the *MRP8^{WT}Csp1^{flox}* controls (Fig 5D). In response to HN878, the decrease was 46%. This result confirms the direct role of neutrophilic Nlrp3 inflammasome activation in lung inflammation. Among leukocytes, 3.2 times fewer neutrophils (Fig 5E) were recruited to the lungs of *MRP8^{Cre+}Csp1^{flox}* than *MRP8^{WT}Csp1^{flox}* mice in response to H37Rv and 6.3 times fewer were recruited in response to HN878. In H37Rv-infected mice, we observed fivefold fewer inflammatory neutrophils in the lungs of *MRP8^{Cre+}Csp1^{flox}* mice than the *MRP8^{WT}Csp1^{flox}* controls and approximately twofold—but not statistically significant—fewer regulatory neutrophils (Fig 5F). On the contrary, in HN878-infected mice, the number of inflammatory neutrophils was 16 times lower in *MRP8^{Cre+}Csp1^{flox}* than *MRP8^{WT}Csp1^{flox}* mice, whereas the number of regulatory neutrophils was only 2.3 times lower (Fig 5F). Thus, the absence of neutrophilic inflammasome activation had a greater impact on inflammatory neutrophils than regulatory neutrophils, which was even more marked in response to HN878 infection. Again, we observed a much lower expression of PD-L1 on the surface of regulatory neutrophils in response to HN878 (MFI 2256) than H37Rv infection (MFI 6158) (Fig S3E). However, the levels were similar in *MRP8^{Cre+}Csp1^{flox}* and *MRP8^{WT}Csp1^{flox}* mice, indicating that the ability of inflammatory neutrophils to produce IL-1β did not have an impact on the PD-L1 expression of regulatory neutrophils. CD4 T-cell numbers were 2.7 times lower for *MRP8^{Cre+}Csp1^{flox}* than *MRP8^{WT}Csp1^{flox}* mice in response to H37Rv and 2.3 times lower in response to HN878 (Fig 5G). There was no statistically significant difference in the recruitment of other CD11b[pos] cells in the lung between *MRP8^{Cre+}Csp1^{flox}* and *MRP8^{WT}Csp1^{flox}* mice in response to H37Rv or HN878 (Fig 5H). Despite the greater impact on cell recruitment on day 21 after infection with HN878, we did not observe morphological differences in lung lesions between *MRP8^{Cre+}Csp1^{flox}* and *MRP8^{WT}Csp1^{flox}* mice (Fig 5I), and the surface area occupied by lesions was not statistically different at that time point (Fig 5J).

## Extremely susceptible IFN-γR^{−/−} mice show dysregulation of both neutrophil subsets

Mendelian inherited susceptibility to mycobacteria involves IFN-γR and its signaling cascade (33). IFN-γR^{−/−} mice are extremely susceptible to Mtb infection, and this is linked to strong recruitment and dysregulated cell death of neutrophils (24). We infected IFN-γR^{−/−} mice with virulent H37Rv or avirulent BCG. We did not infect these extremely susceptible animals with hypervirulent HN878 for ethical reasons. As we did not observe any clinical condition in IFN-γR^{−/−} mice infected with BCG (data not shown), we did not pursue neutrophil analysis in these animals. 3 wk after infection with H37Rv, we observed macroscopic lesions in the lungs and livers of IFN-γR^{−/−} mice that were not seen in their WT counterparts (data not shown). As previously reported by Nandi and Behar (24), we observed more sustained viability of neutrophils in Mtb IFN-γR^{−/−} mice as compared to the wild type, and this was true for both subsets (Fig S4A). The lungs of IFN-γR^{−/−} mice showed 2.3 times more CFUs than those of WT mice (Fig 6A). In accordance with the high number of macroscopic lesions, histological analysis of the lungs of IFN-γR^{−/−}-infected mice

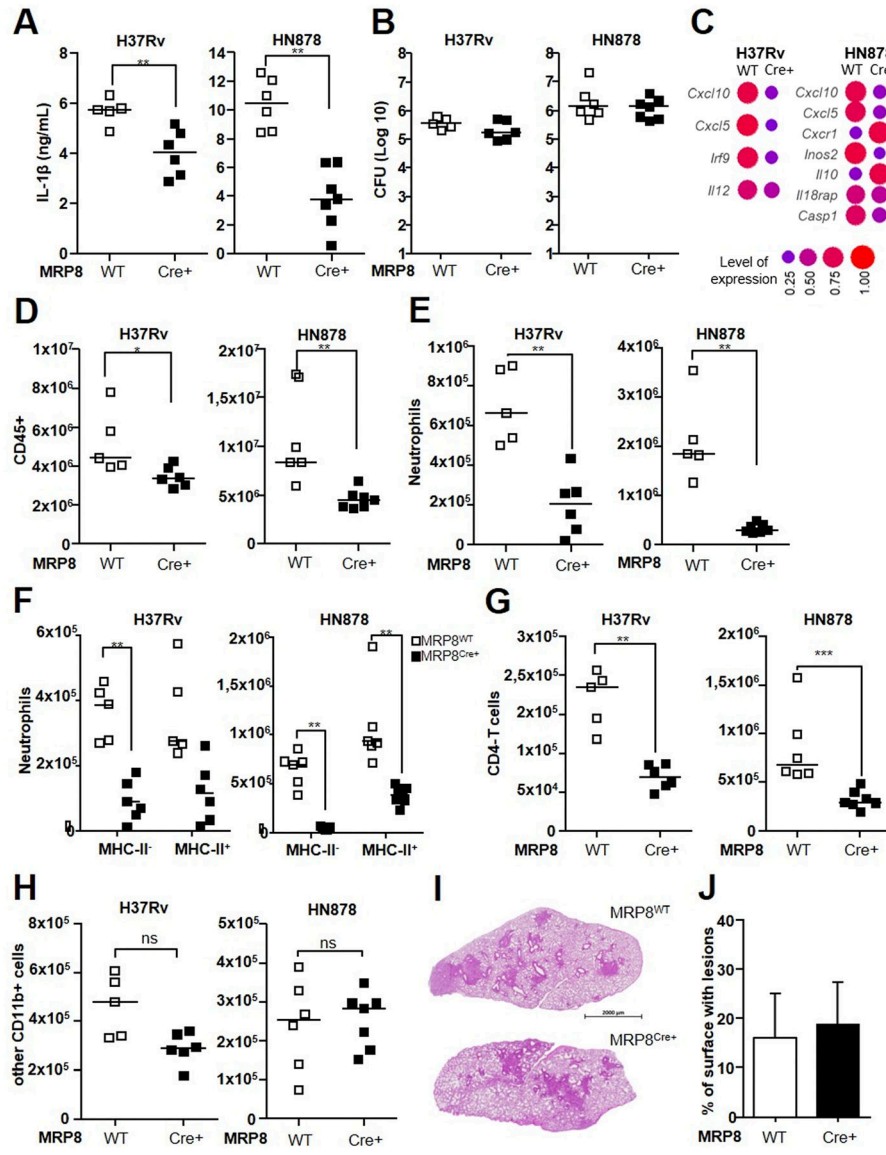

**Figure 5. Caspase-dependent production of IL-1β by inflammatory neutrophils sustains lung inflammation.**
**(A, B, C, D, E, F, G, H, I)** *MRP8^{Cre+}Csp1^{flox}* or *MRP8^{WT}Csp1^{flox}* mice were infected with H37Rv or HN878 and the lungs harvested at 21 d. **(A)** IL-1β production quantified by ELISA in lung tissue homogenates. **(B)** Number ($Log_{10}$) of CFUs for each animal. **(C)** Differential gene expression of a panel of 48 genes between *MRP8^{Cre+}Csp1^{flox}* and *MRP8^{WT}Csp1^{flox}* mice infected with H37Rv or HN878. The dot plot represents the normalized expression of significantly deregulated genes expressed as a normalized rate to compare the two groups. **(D, E, F, G, H)** Cells were analyzed by flow cytometry, and the number of (D) CD45+ total leukocytes, (E) Ly-6G+ total neutrophils, (F) MHC-II− and MHC-II+ neutrophil subsets, (G) CD4+ T cells, and (H) other CD11b+ cells was compared between *MRP8^{Cre+}Csp1^{flox}* and *MRP8^{WT}Csp1^{flox}* mice. **(I)** Representative section of hematoxylin/eosin lung staining for each group. **(J)** Total lung surface occupied by lesions in lung sections from *MRP8^{Cre+}Csp1^{flox}* or *MRP8^{WT}Csp1^{flox}* mice. Data are presented as individual data points and medians. **(A, B, C, D, E, F, G, H, I)** n = 5–7 mice per group; (A, B, D, E, F, G, H, I, J) two independent experiments; (C) one experiment. *P < 0.05, **P < 0.01, and ***P < 0.001 by the Mann–Whitney test (A, B, C, D, E, G, H) and the nonparametric Fisher–Pitman permutation test (F).

showed extensive, disorganized inflammatory cell infiltrates (Fig 6B). The total surface occupied by lesions was 2.6 times higher for the IFN-γR^{−/−} than WT mice (Fig 6C). As expected (24), we observed twofold greater recruitment of total leukocytes to the lungs of IFN-γR^{−/−} than WT mice (Fig 6D). This difference was mainly due to total neutrophils, which were 6.5 times more abundant in IFN-γR^{−/−} than WT mice. The number of CD4+ T cells was also twofold higher in the lungs of IFN-γR^{−/−} mice, whereas there was no difference in the number of CD8+ T cells (Fig S4B). Of note, inflammatory neutrophils represented 70% and regulatory neutrophils represented 30% of the total neutrophil influx in IFN-γR^{−/−} mice (Fig 6D), whereas the neutrophil influx in WT controls was balanced between the inflammatory (41%) and regulatory (59%) subsets. The threefold higher level of IL-1β detected in the lungs of IFN-γR^{−/−} than WT mice (Fig 6E) was consistent with the higher influx of inflammatory neutrophils. Higher inflammation was also indicated by the presence of 2.6 times more TNF (Fig 6F) and 1.7

times more IL-6 (Fig 6G) in the lungs of IFN-γR^{−/−} than WT mice. The lung tissue from the two mouse strains showed highly different transcriptional profiles in response to H37Rv infection (Fig 6H). Genes such as *Cxcl1*, *Cxcr1*, *Cxcr2*, *Mmp7*, *Mmp8*, *Mmp9*, *Mpo*, and *S100a8* were more highly expressed in IFN-γR^{−/−} than WT mice (Fig 6I). Many genes, such as *Ilr1* and *Ilr2*, which are highly expressed during inflammation, including by the neutrophils themselves, were also more highly expressed in IFN-γR^{−/−} than WT mice. In contrast, the expression of type I IFN–related genes was higher in WT than IFN-γR^{−/−} mice.

### Hyperinflammation in IFN-γR^{−/−} mice is relieved by IFN-γR+ regulatory neutrophils

The genes for which the expression was higher in the lungs of WT than those of IFN-γR^{−/−} mice in response to H37Rv infection included *Mhc-II*, *CD274*, *Cd86*, and *Cd40* (Fig 6I), which are all involved

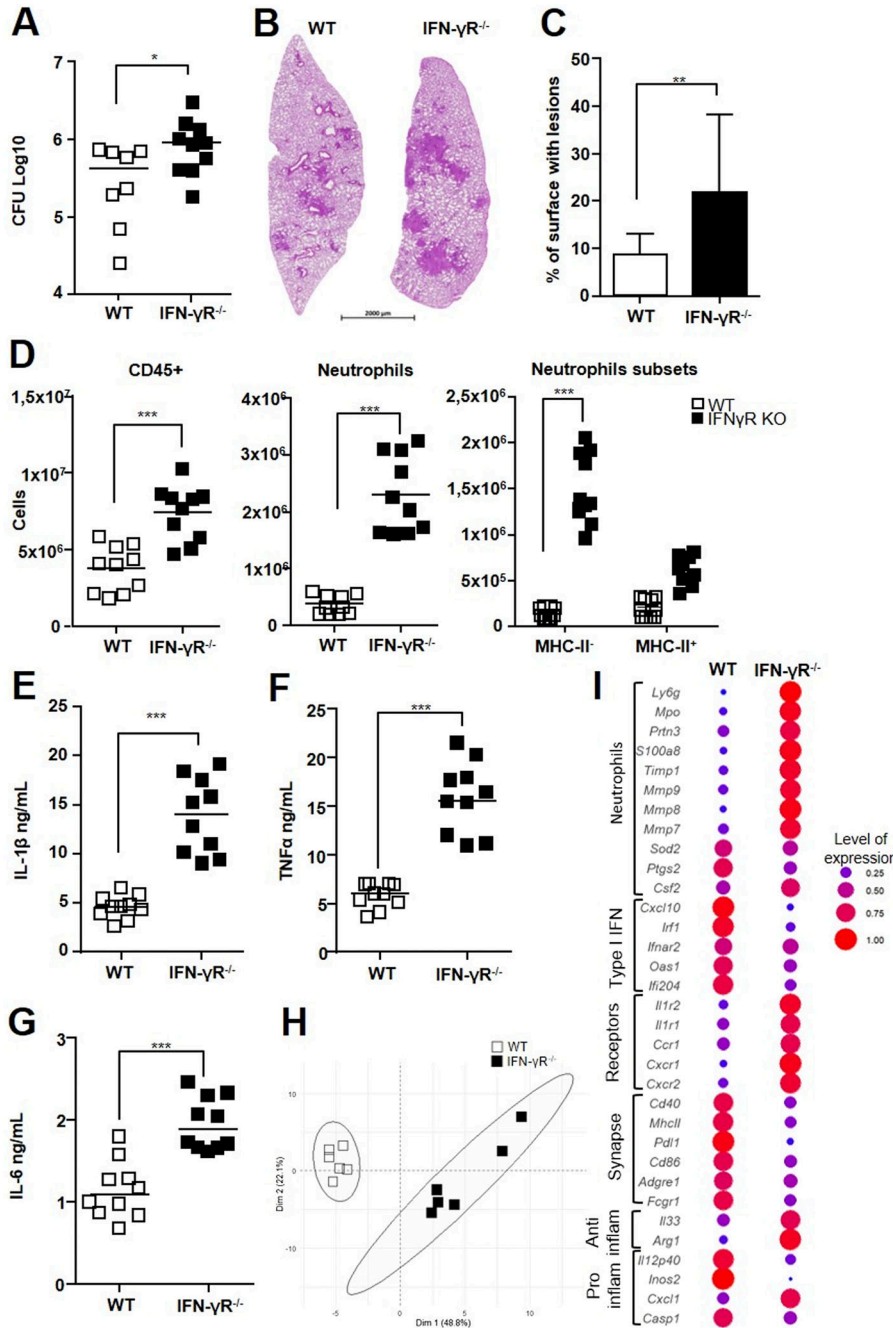

**Figure 6. Extremely susceptible IFN-γR⁻/⁻ mice display dysregulation of both neutrophil subsets.**
C56BL/6 WT or IFN-γR⁻/⁻ mice were infected with H37Rv and the lungs harvested on day 21 for analysis. **(A)** Number (Log₁₀) of CFUs for each animal. **(B, C)** Representative section of hematoxylin/eosin lung staining for each group and **(C)** mean percentage of lung surface occupied by lesions. **(D)** Cells were analyzed by flow cytometry, and the number of CD45⁺ total leukocytes, Ly-6G⁺ total neutrophils, and MHC-II⁻ and MHC-II⁺ neutrophil subsets was compared between the two groups of mice. **(E, F, G)** Cytokine production was analyzed by ELISA in lung tissue homogenates. **(H, I)** Gene expression of a panel of 48 genes in the lungs was assessed by Fluidigm BioMark. **(H)** mRNA expression was normalized to the expression of three housekeeping genes and to the uninfected group to calculate the ΔΔCt. Principal component analysis was performed on the ΔΔCt values. The two first dimensions of the Principal component analysis plot are depicted. **(I)** Dot plot represents the normalized expression of significantly deregulated genes expressed as a normalized rate to compare the C56BL/6 WT and IFN-γR⁻/⁻ mice. **(A, B, C, D, E, F, G, H, I)** Data are presented as the mean of n = 5–7 mice per group; (A, B, C, D, E, F, G) pooled data from two independent experiments; (G, H) analysis of one experiment. Graphs are presented as individual data points and medians ((C), median with range). **(G)** Data are presented as the mean of n = 6 mice per group. ∗P < 0.05, ∗∗P < 0.01, and ∗∗∗P < 0.001 by the Mann–Whitney test (A, B, C, D, E, F, G, H, I) and the non-parametric Fisher–Pitman permutation test ((D), neutrophil subsets).

in the synapse between antigen-presenting cells and T cells. As these mice showed a hyperinflammatory profile, we investigated the impact of the IFN-γR on regulatory neutrophils. The level of MHC-II expression was not affected by the absence of the IFN-γR (Fig 7A). However, regulatory neutrophils from IFN-γR⁻/⁻ mice lost PD-L1 surface expression, showing levels similar to those of inflammatory MHC-II⁻ neutrophils from WT animals (Fig 7B). Moreover, the proportion of MHC-II⁺ neutrophils that expressed low levels of PD-L1 in IFN-γR⁻/⁻ mice dropped to 30%, whereas 90% of MHC-II⁺ neutrophils highly expressed PD-L1 in WT animals (Fig 7C). We

enriched lung regulatory neutrophils from IFN-γR⁻/⁻ or WT mice 21 d after H37Rv infection by magnetic sorting. Strikingly, IFN-γR⁻/⁻ regulatory neutrophils completely lost the ability to suppress OT-II cell proliferation (Fig 7D) and IFN-γ production ex vivo (Fig 7E), showing that the control exerted by regulatory neutrophils on T cells is dependent on the IFN-γR. Thus, we hypothesized that lethal inflammation in Mtb-infected IFN-γR⁻/⁻ mice was linked to strong recruitment of inflammatory neutrophils and less efficient control of inflammation by regulatory neutrophils because of PD-L1 down-regulation. We tested this hypothesis by harvesting PD-L1ʰⁱ

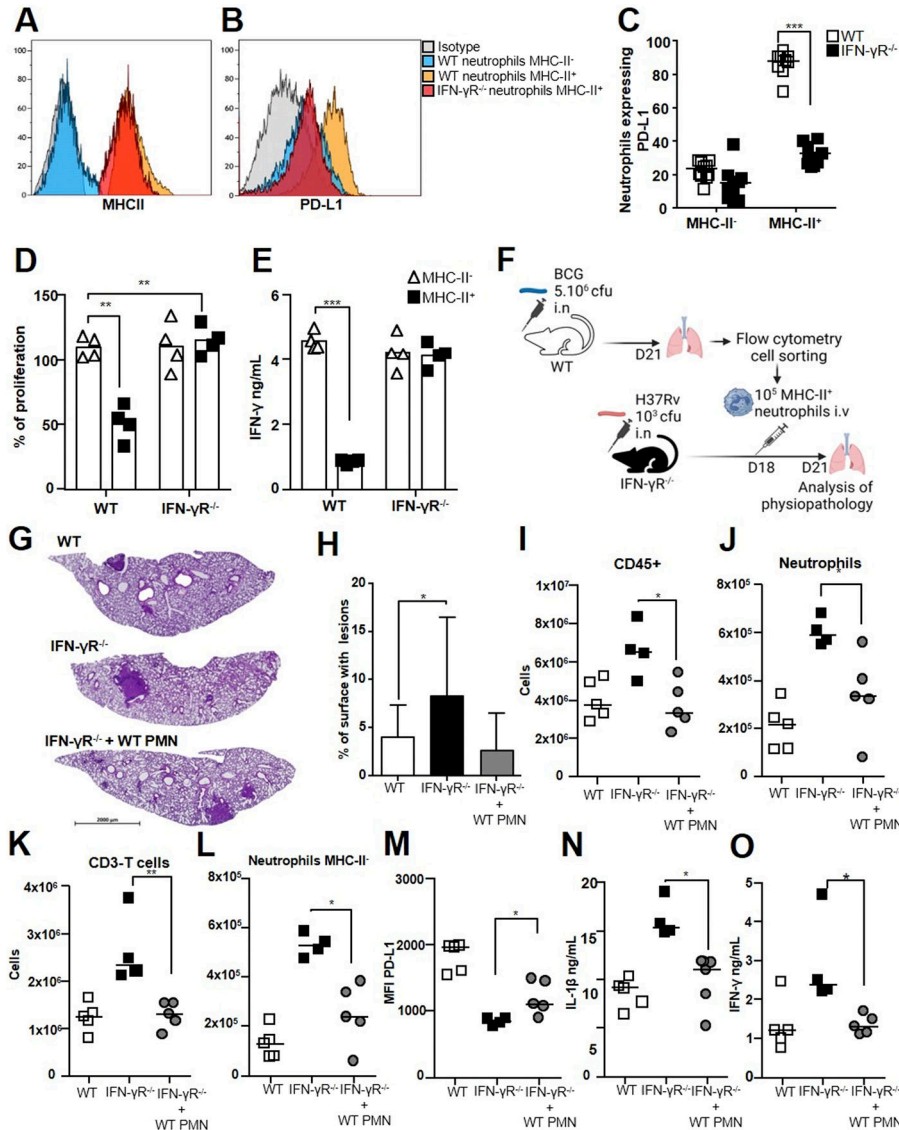

**Figure 7. Hyperinflammation in IFN-γR⁻/⁻ mice is relieved by IFN-γR⁺ regulatory neutrophils.**
C56BL/6 WT or IFN-γR⁻/⁻ mice were infected with H37Rv and the lungs harvested on day 21 for analysis. **(A, B)** Neutrophil subsets were analyzed by flow cytometry for (A) MHC-II and (B) PD-L1 surface expression. **(C)** Percentage of neutrophils expressing PD-L1 on the surface among the two MHC-II⁻ and MHC-II⁺ neutrophil subsets in the two mouse groups. **(D, E)** On day 21, MHC-II⁻ and MHC-II⁺ neutrophil subsets were enriched by magnetic beads from the lungs of the two groups of mice and mixed with OT-II cells. **(D, E)** Percentage of OT-II splenocyte proliferation and (E) IFN-γ production in the presence of each neutrophil subset calculated based on the response of OT-II splenocytes to the Ova peptide only. **(F)** Schematic representation of the transfer of [MHC-II⁺, PD-L1ʰⁱ] regulatory neutrophils purified from the lungs of BCG-infected WT mice into IFN-γR⁻/⁻ mice infected with H37Rv 18 d before. The three H37Rv-infected groups harvested on day 21 were WT control mice and IFN-γR⁻/⁻ mice that were mock-treated or to which WT regulatory neutrophils were transferred. **(G, H)** Representative section of hematoxylin/eosin lung staining for each group and (H) the percentage of lung surface occupied by lesions analyzed. **(I, J, K, L, M, N, O)** Cells were analyzed by flow cytometry to determine the number of (I) CD45⁺ total leukocytes, (J) Ly-6G⁺ total neutrophils, (K) T cells, and (L) MHC-II⁻ inflammatory neutrophils. **(M)** Comparison of the PD-L1 mean fluorescence intensity on MHC-II⁺ regulatory neutrophils between the three groups analyzed by flow cytometry. **(N, O)** IL-1β and (O) IFN-γ production measured in lung tissue homogenates by ELISA. **(A, B, C, D, E, G, H, I, J, K, L, M, N, O)** Histograms are representative of two independent experiments, n = 6 or 4; (C) pooled data from two independent experiments, n = 10; (D, E) pooled data from two independent experiments (n = 4); (G, H, I, J, K, L, M, N, O) data from one experiment, n = 4–5. Data are presented as individual data points and medians ((H), with range). ∗P < 0.05, ∗∗P < 0.01, and ∗∗∗P < 0.001 by the non-parametric Fisher–Pitman permutation test (C, D, E) and the Mann–Whitney test (G, H, I, J, K, L, M, N, O).

regulatory neutrophils from BCG-infected WT mice and transferring them into H37Rv-infected IFN-γR⁻/⁻ mice on day 18 post-infection. On day 21, we euthanized these mice, as well as the two control groups, H37Rv-infected WT and mock-treated IFN-γR⁻/⁻ mice (Fig 7F), and assessed the TB pathophysiology in the lungs. Again, we observed more sustained viability of both subsets of neutrophils in Mtb IFN-γR⁻/⁻ mice as compared to the wild type (24) (Fig S5A). However, viability of CD3 T cells was not different among the groups (Fig S5A). We did not observe any significative differences in CFU counts between the groups at this time point (Fig S5B). However, transfer of PD-L1ʰⁱ regulatory neutrophils relieved inflammation in the lung tissue from IFN-γR⁻/⁻ mice (Fig 7G), although the difference in total lung surface occupied by lesions among mock-treated and regulatory neutrophil transferees did not reach statistical significance (Fig 7H). Nonetheless, the dampening of inflammation was also indicated by a significant reduction in total leukocyte numbers (Fig 7I), in particular, those of neutrophils (Fig 7J) and T cells (Fig 7K),

including both CD8⁺ (Fig S5C) and CD4⁺ T cells (Fig S5D). The transfer of regulatory neutrophils to IFN-γR⁻/⁻-infected mice dampened the strong recruitment of inflammatory neutrophils observed in mock-treated animals (Fig 7L), and the PD-L1 MFI of MHC-II⁺ neutrophils increased significantly (Fig 7M). Consistent with these results, we measured 1.4-fold less IL-1β production in the lung tissue from transferees than mock-treated animals (Fig 7N). IFN-γ levels were also 1.8-fold lower (Fig 7O), and those of TNF remained unchanged at this time point (Fig S5E).

## Discussion

TB pathophysiology in the lung is characterized by a delicate balance between pro- and anti-inflammatory mechanisms controlled by both the host and bacilli. Neutrophils are widely recognized as "bad guys" in TB, playing key roles in lung destruction

(12, 15, 34). However, we propose here a more subtle definition of neutrophils based on the discovery of two distinct subsets, recruited to the lungs in response to mycobacterial infection. On the one hand, the inflammatory neutrophil subset produces caspase-1–dependent IL-1β and acts as an accelerator of local inflammation in response to virulent mycobacteria by maintaining a vicious circle of inflammatory neutrophils and CD4 T cells. On the other hand, the regulatory neutrophil subset dampens inflammation by blocking T-cell proliferation and IFN-γ production. The IFN-γR–dependent expression of PD-L1 on regulatory neutrophils is critical for the braking function. Regulatory neutrophils are less affected than inflammatory neutrophils by the absence of neutrophil-derived IL-1β, suggesting differential regulation mechanisms. Our data shed new light on the role of neutrophils in TB, and we propose that these two subsets are involved in a "brake/accelerator" inflammation circuit in the lungs during TB infection. Moreover, the two brake and accelerator pedals could represent a means for hypervirulent Mtb strains to manipulate the host's immune system and establish a successful infection (Graphical Abstract).

IL-1β is a double-edged sword during TB infection that must be tightly controlled. It is involved in strong neutrophil recruitment to the lungs during severe TB (12, 35, 36) and is a target for host-directed therapies (37). Neutrophils produce IL-1β during Mtb infections, and we demonstrated here that caspase-1–dependent cleavage of pro-IL-1β occurs in neutrophils, in addition to protease-dependent mechanisms (38). We observed that avirulent BCG could trigger caspase-1–dependent IL-1β production by neutrophils in vitro, showing that the major virulence factor ESAT-6, which is present in Mtb and absent from BCG, was dispensable. However, in vivo, caspase-1–dependent IL-1β production by neutrophils was only induced by Mtb and not BCG, indicating that other regulatory pathways are involved in inflammasome activation in the lungs. 3 wk after Mtb infection of mice bearing caspase-1–defective neutrophils, we observed a 30% to 64% reduction of IL-1β levels in the lungs depending on the virulence of the strain and a coincident 3.2- to 6.3-fold reduction in total neutrophil recruitment, underlining the importance of the caspase-1–dependent pathway for the inflammatory loop involving neutrophils in the lung. In our study, restricted to one time point corresponding to early orchestration of the adaptive T-cell response in the lungs (16, 39), we observed that caspase-dependent IL-1β production mainly affected the recruitment of inflammatory neutrophils and CD4 T cells. The link between IFN-γ–producing CD4 T cells and excessive neutrophilia during clinical manifestations of TB has been clearly established (40), and we observed high recruitment of IL-1β–producing inflammatory neutrophils in IFN-γR⁻/⁻ mice that correlated with elevated levels of CD4 T cells and IFN-γ and highly lesioned lungs. Thus, IL-1β–producing inflammatory neutrophils are more involved than regulatory neutrophils in severe TB in this model. Although we did not observe a major impact of caspase-dependent IL-1β production by inflammatory neutrophils on control of the bacilli or lesion formation at the early time point of our studies, we believe that other time points should be examined in *MRP8^Cre+^Csp1^flox^* mice to gain a better understanding of caspase-1– versus protease-dependent mechanisms of IL-1β production by neutrophils.

In highly susceptible IFN-γR⁻/⁻ mice, we observed that the strong neutrophilia was driven by two paths, dysregulated recruitment of IL-1β–producing inflammatory neutrophils and dysfunction of PD-L1^hi regulatory neutrophils, which could be alleviated by the transfer of competent WT regulatory neutrophils. The immune checkpoint inhibitor PD-L1 was critical for the function of regulatory neutrophils, akin to neutrophils present in cancer (41), which foster immune suppression in hepatocellular carcinoma (42, 43) and gastric cancer (44). Competent PD-L1^hi regulatory neutrophils were also recruited to the lungs in response to avirulent BCG infection, indicating that the acquisition of this function did not fully depend on mycobacterial virulence. Recently, PD-L1⁺ neutrophils were described in two acute disorders, sepsis (45, 46) and cutaneous burn injury (47), as well as during chronic infections in cutaneous (48) or visceral (49) leishmaniasis. During *Candida albicans* infection, PD-L1⁺ neutrophils decrease antifungal immunity by retaining the pool of microbiocidal neutrophils in the bone marrow (50). We found that the IFN-γR was required for PD-L1 expression and suppression of CD4 T cells. Similarly, human (51) and mouse neutrophils need to be exposed to IFN-γ to express PD-L1 and suppress T cells during endotoxemia (45). Moreover, we found that the transfer of WT regulatory neutrophils that expressed PD-L1 into Mtb-infected IFN-γR⁻/⁻ mice alleviated exuberant lung neutrophilia in these extremely susceptible animals.

The tremendous success of Mtb as a pathogen can be explained by its co-evolution with that of the host. Strains from the Beijing family are among the most successful, as demonstrated by their global distribution and the recurrent outbreaks they cause (52). This success is partially due to their exquisite ability to manipulate the host's immune system. The peculiar cell wall composition allows the Beijing strains to immunosuppress the innate immune response (53), especially in microaerophilic or anaerobic environments (54), such as that encountered in the granuloma. Here, we confirm the hypervirulence of the Beijing prototype strain HN878 in C57BL/6 mice, with neutrophil-driven lung inflammation (28, 29). A neutrophil-driven type I IFN response has been shown to lead to a poor prognosis for TB patients (55) and mice (56). Our data, restricted to one time point in C57BL/6 mice, indicate better induction of the type I IFN pathway in the lungs by the less virulent H37Rv strain than hypervirulent HN878. However, unlike H37Rv, HN878 was able to fuel the neutrophil influx toward recruitment of the regulatory subset, with diminished PD-L1 expression. Mtb Beijing strains induce regulatory T-cell expansion (57, 58) better than laboratory-adapted strains. They also favor recruitment of myeloid-derived suppressor cells producing IL-10, which could limit excessive lung damage (28). Of note, we also observed the higher expression of *Il10* and *Arg1* in the lungs of mice infected with HN878 than those infected with H37Rv. Our most striking finding was the ability of HN878 to recruit a neutrophil compartment biased toward regulatory neutrophils, which expressed threefold to fivefold less PD-L1 on their surface than less virulent laboratory-adapted strains. We observed a similar difference at the transcriptional level in lung tissue. As PD-L1 is widely expressed by both myeloid and non-hematopoietic cells (59), it is possible that control of this important immune checkpoint by HN878 occurs at several levels at the site of infection. Further studies are required to better dissect the mechanisms used by diverse Mtb strains to finely tune PD-L1

expression and of how they relate to the functional consequences of infection. We believe that regulatory neutrophils acting as a "brake pedal" represent yet another weapon in the arsenal of Beijing strains to manipulate the immune system and establish successful infection.

Our study had two principal limitations. First, the function of regulatory neutrophils was not assessed in TB patients. Although PD-L1[+] neutrophils have been found in TB patients ([60]), this subset is yet to be investigated in the lungs of humans and non-human primates. Second, although the C57BL/6 mouse model is the most widely used because it allows mechanistic studies in genetically modified mice, granulomas are not well formed in the lungs in response to Mtb infection in this model. Therefore, our next step will be to examine the contribution of the two neutrophil subsets to granuloma formation in C3HeB/FeJ mice ([61]).

In conclusion, our results add a new layer of complexity to the multiple functions exerted by different neutrophil subsets during TB and emphasize their key role as partners of the immune response. Inflammatory neutrophils are certainly "foes," worsening TB pathogenesis in the lung. It is yet to be determined whether regulatory neutrophils are "friends" and associated with a good prognosis for TB patients. Recent demonstration of the importance of the PD-1/PD-L1 axis in the control of TB ([6]) and the data we report here on regulatory PD-L1[hi] neutrophils open new avenues to explore the role of this subset in the granuloma microenvironment in humans. As neutrophils, in general, are an important target for the development of new host-directed therapies ([62], [63]), it is urgent to reconsider the complexity of these cells to better target pharmaceutical and immune interventions in TB.

# Materials and Methods

## Experimental design and justification of the sample size

Mice were bred at the specific pathogen-free animal facility Plateforme Infectiologie Experimentale (PFIE, U1277, INRAE, Centre Val de Loire). 1 wk before in vivo experiments, mice were moved from the specific pathogen-free breeding area to the ABSL3 area to acclimate them. For infections with Mtb, mice were placed in biological safety cabinets. Mice were housed in groups of four to five per cage and randomly distributed. $MRP8^{Cre+}Csp1^{flox}$ and $MRP8^{WT}Csp1^{flox}$ mice were littermates. Groups always contained at least four individuals to allow statistical assessment of the data using non-parametric Mann–Whitney tests. Results were not blinded for analysis except for the RNAseq analysis. The number of biological replicates and experiments is indicated in the figure legends.

## Ethics statement

Experimental protocols complied with French law (Décret: 2001–464 29/05/01) and European directive 2010/63/UE for the care and use of laboratory animals and were carried out under Authorization for Experimentation on Laboratory Animals Number D-37-175-3 (Animal facility UE-PFIE, INRAE Centre Val de Loire). Animal protocols were approved by both the "Val de Loire" Ethics Committee for Animal Experimentation and the French Minister of Higher Education, Research and Innovation. They were registered with the French National Committee for Animal Experimentation under N° APAFIS #35838-2022031011022458.v5.

## Mice

$MRP8^{Cre+}Csp1^{flox}$ mice have been previously described ([25]). Because the introduction of the *cre* gene encoding the recombinase under control of the MRP8 promoter in both alleles of the C57BL/6 mouse chromosome was lethal, we bred and screened mice to obtain $MRP8^{Cre+}Csp1^{flox}$ mice in which one allele carried the CRE recombinase, whereas the other did not. In these animals, the expression of the recombinase under the MRP8 promoter induced excision of the *Csp1*-encoding genes in 100% of the neutrophils. Control $MRP8^{WT}Csp1^{flox}$ mice did not carry the recombinase under the control of the MRP8 promoter, and neutrophils were able to cleave pro-IL-1β. All mice were bred in-house, except OT-II mice, which were purchased from Janvier Biolabs.

## Bacterial strains and growth conditions

All mycobacterial strains (*Mycobacterium bovis* BCG strain WT 1173P2 Pasteur, Mtb strains H37Rv and HN878) were cultivated for 12 d in Middlebrook 7H9 broth (Becton Dickinson) supplemented with 10% BBL Middlebrook ADC enrichment (Becton Dickinson) and 0.05% Tween-80 (Sigma-Aldrich), aliquoted, and frozen at −80°C in 7H9 medium containing 10% glycerol. Bacterial suspensions for infection were prepared in PBS from quantified glycerol stock solutions. To enumerate the number of CFUs from the middle right lung lobe, tissue was homogenized, and serial dilutions were plated on supplemented 7H11 plates as previously described ([16]).

## Intranasal infection and treatments

Mice anesthetized by i.p. injection of a ketamine/xylazine cocktail received $5 \times 10^6$ CFUs of BCG Pasteur or $10^3$ CFUs of Mtb H37Rv or HN878 in 20 µl in each nostril. For total neutrophil depletion experiments, C57BL/6 mice received 200 µg anti-Ly-6G antibody (clone 1A8; BioLegend) via the i.p. route on days 15, 17, and 19 after BCG inoculation. Control mice were injected with the same quantity of IgG2b Ab (BioLegend). For neutrophil transfer experiments, MHC-II[+] neutrophils were isolated from the lungs of BCG-infected mice harvested on day 21. IFN-γR[−/−] and control C57BL/6 mice that were infected with H37Rv 18 d before were injected i.v. with $1.5 \times 10^5$ MHC-II[+] neutrophils. Mice were euthanized on day 21 for analysis of the lungs. All mice were euthanized by pentobarbital administration at the time post-infection as indicated in the figure legends.

## Preparation of neutrophils and macrophages from bone marrow

Femurs were harvested from 6-wk-old mice (WT, $MRP8^{WT}Csp1^{flox}$, and $MRP8^{Cre+}Csp1^{flox}$) bred at the PFIE animal facility. Femurs from $Aim2^{−/−}$, $Gsdmd^{−/−}$, $Nlrp3^{−/−}$, and $Csp1/11^{−/−}$ mice were kindly donated by Valérie Quesniaux (INEM, UMR7355, CNRS, University of Orleans, France) and those of $Csp1^{−/−}$ mice by Sergio Costa

(Universidade Federal de Minas Gerais, Belo Horizonte, Brazil). Neutrophils were directly purified from bone marrow by anti-Ly-6G magnetic positive selection (Miltenyi Biotec), as previously described (64). Neutrophils of > 95% purity were obtained as assessed by microscopy after May–Grünwald–Giemsa staining. Viability by trypan blue exclusion was 98%. MPs were obtained after culturing with 30% L929 cell-conditioned medium as a source of macrophage colony-stimulating factor. Cells used on day 10 for infectivity and cytokine assays were suspended in complete medium without antibiotics, as previously described (64). Macrophages (1 × 10$^5$/well) or neutrophils (1 × 10$^6$/well) were plated in P96 plates, infected overnight with BCG at an MOI of 10 or Mtb at an MOI of 1, or stimulated overnight with 100 ng LPS (from *E. Coli* 011: B4; Sigma-Aldrich) and 10 $\mu$M nigericin sodium salt (Sigma-Aldrich) added 1 h before harvesting the cells and supernatants.

## Lung cell preparation and flow cytometry

Briefly, euthanized mice were perfused with PBS and the left lung lobes digested for 1 h with collagenase D (1.5 mg/ml, Roche) and DNase A (40 U/ml, Roche) before filtering cells through a 100-$\mu$M nylon cell strainer (BD Falcon). For extracellular staining, cells were incubated for 20 min with 2% total mouse serum and labeled in PBS supplemented with 5% FCS and 0.1% total mouse serum with viability dye (eBioscience) and antibodies against the surface markers, all from BD Biosciences (listed in Table S1). Intracellular mature IL-1$\beta$ production was measured using anti-IL-1$\beta$ biotin-conjugated antibody (Rockland) after treatment for 2 h at 37°C with 5 $\mu$g/ml brefeldin A (Sigma-Aldrich). Cells were washed and fixed with BD cell fix diluted 4X in PBS. Data were acquired on an LSRFortessa X-20 flow cytometer (Becton Dickinson) and the results analyzed using Kaluza software (Beckman Coulter).

Lung regulatory or inflammatory neutrophils were prepared from the lungs of C57BL/6 mice on day 21 post-infection with Mtb or BCG. For Mtb, lungs were digested and the neutrophils isolated by magnetic bead selection using the untouched neutrophil isolation kit according to the manufacturer's instructions (Miltenyi). MHC-II–positive magnetic bead selection was performed on the unlabeled neutrophil-rich fraction. MHC-II$^+$ regulatory neutrophils were separated from MHC-II$^-$ inflammatory neutrophils using anti-MHC-II PE-conjugated Ab (BD Biosciences) and anti-PE beads (Miltenyi). Neutrophil subset purity was between 70 and 81% and viability superior to 95%. Neutrophils were recovered in complete medium and immediately processed for suppressive activity assay.

For BCG, after excluding dead cells, total neutrophils [CD11b$^+$, Ly-6C$^+$, Ly-6G$^+$], classic neutrophils [CD11b$^+$, Ly-6C$^+$, Ly-6G$^+$, MHC-II$^-$], or regulatory neutrophils [CD11b$^+$, Ly-6C$^+$, Ly-6G$^+$, MHC-II$^+$] were sorted on a MoFlo Astrios EQ high-speed cell sorter (Beckman Coulter) as previously described (18). Neutrophil subsets of > 99% purity were obtained in each fraction. Neutrophils were recovered in complete medium and immediately processed for single-cell RNAseq analysis (total neutrophils) or transcriptomic analysis or suppressive activity assay or neutrophil transfer (neutrophil subsets).

## Measurement of T-cell suppressive activity of neutrophils

The T-cell suppressive activity of neutrophils was measured as previously published (18). Briefly, total splenocytes from OT-II mice were collected, homogenized to single-cell suspensions through nylon screens, and resuspended in RPMI medium (Gibco) supplemented with 10% decomplemented fetal bovine serum (Gibco), 2 mM L-glutamine (Gibco), 100 U penicillin, and 100 $\mu$g/ml streptomycin (Gibco). Then, 10$^5$ cells/well were distributed in a 96-well round-bottom plate (BD Falcon). OT-II splenocyte proliferation was induced by the addition of 2 $\mu$g/ml of the OVA peptide 323-339 (PolyPeptide Group). Purified neutrophils were added to the cultured splenocytes at a ratio of 1:10 in a final volume of 200 $\mu$l. Wells without neutrophils were used as a reference for maximal proliferation. Cell proliferation was quantified after 3 d of culture using CyQUANT Cell Proliferation Assay tests (Thermo Fisher Scientific) according to the manufacturer's instructions. The role of PD-L1 in the suppression mechanism was assessed by incubating sorted neutrophils for 1 h with 50 $\mu$g/ml anti-PD-L1 Ab (Tecentriq, atezolizumab) or a human IgG1 isotype control before mixing with OT-II splenocytes. Cell proliferation was quantified after 3 d of culture using CyQUANT Cell Proliferation Assay tests (Thermo Fisher Scientific) according to the manufacturer's instructions.

## Medium-throughput and single-cell RNA sequencing of neutrophils

For medium-throughput analysis of gene expression in neutrophils, total RNA was extracted from FACS-sorted neutrophils from the right accessory lung lobe homogenized using Lysing Matrix D tubes from MP Biomedicals and Precellys using a NucleoSpin RNA kit with DNase treatment (Macherey-Nagel). Total RNA was reverse-transcribed using an iScript Reverse Transcriptase mix (Bio-Rad) and gene expression assessed using BioMark HD (Fluidigm) according to the manufacturer's instructions or LightCycler 480 Real-Time PCR System (Roche). The annealing temperature was 62°C. All primers are listed in Table S2. Data were analyzed using Fluidigm RealTime PCR software or Lc480 software to determine the cycle threshold (Ct) values. Messenger RNA (mRNA) expression was normalized against the mean expression of three housekeeping genes for each sample to obtain the ΔCt value. Infected samples were normalized against uninfected samples (ΔΔCt). Relative gene expression was calculated according to the formula RQ = $2^{-\Delta\Delta Ct}$. Dot plots were created using RStudio for differentially expressed genes between the two groups (Mann–Whitney) by normalizing the fold change of each group to the total fold expression for each gene (normalized rate = fold change group 1/fold change group 1 + fold change group 2). This normalized rate is represented as spot plots when the transcriptomes of two groups are compared.

For single-cell analysis, viable total Ly-6G$^+$ neutrophils were sorted using a MoFlo Astrios high-speed cell sorter. Within 1 h after sorting, cells were encapsulated with barcoded Single Cell 3' v3.1 Gel Beads and a Master Mix to form a Gel Beads-in-emulsion using the 10X Genomics Chromium technology. ~12,000 cells were used. The Single Cell 3' libraries were then generated as recommended by the manufacturer (10x Genomics). The libraries were equimolarly pooled and sequenced (paired-end sequencing) using one lane of

an Illumina NovaSeq 6000 device (IntegraGen), yielding a total of 640 million reads. Raw sequencing data are available under the following BioProject accession number PRJNA1026083. Fastq files were analyzed, and the sequences were aligned against those of the *Mus musculus* genome mm10 (GRCm38, release 98) using the cellranger count pipeline of CellRanger software (v6.0.2). Downstream analyses were performed using R (v4.3.0), RStudio, and the following packages: Seurat (v4.4.0), SingleR (v2.2.0), celldex (v1.10.1). Quality controls first included empty droplets and doublet removal. Then, only droplets with at least 100 features and 1,000 counts were retained. Normalization was done using the LogNormalize method and the 3,000 most variable features.

The dimension reduction method was performed by principal component analysis retaining the first nine principal components with a resolution of 0.8 to identify the clusters with FindClusters.

The cell-type inference was performed using SingleR and the Immunologic Genome Project database retrieved through celldex package.

### ELISA

Cell culture supernatants or right caudal lung lobe tissues, homogenized as above and supplemented with anti-proteases (ROCHE), were passed through 0.2-$\mu$m filters and either processed immediately or frozen at −20°C. Cytokine levels were measured by ELISA using kits (R&D Systems) according to the manufacturer's instructions. Absorbance was measured on a Multiskan FC plate reader (Thermo Fisher Scientific).

### Histology

The right cranial lung lobe was fixed in 4% PFA for 48 h. Subsequently, the tissue was dehydrated and stored in 70% ethanol before being embedded in paraffin. Five-micrometer sections were cut and stained with hematoxylin and eosin (H&E) using a slide stainer (ST5020; Leica Biosystems). All slides were scanned on a slide scanner (AxioScan Z1; Zeiss). Morphological analyses were performed using QuPath software ([65]; available at https://qupath.github.io/), version 0.4. Briefly, airway lesions were quantified using a semi-automated macro. The total area of tissue was automatically measured using a threshold, and the lesions were blindly measured manually for all slides. Data for each mouse consist of the mean of eight sections, cut every 100 $\mu$m, to accurately represent the whole lung.

### Western blots

Neutrophils ($5 \times 10^6$) s were seeded in six-well plates in Opti-MEM/GlutaMAX medium (Gibco) at 37°C. Cells were infected for 5 h with BCG at an MOI of 10 or 20 or stimulated with 500 ng LPS from *E. coli* 011: B4 (Sigma-Aldrich) and 10 $\mu$M nigericin sodium salt (Sigma-Aldrich) added 45 min before the end of the incubation. Then, 1.5 mM AEBSF anti-protease (Sigma-Aldrich) was added to the wells, and the supernatants were clarified by centrifugation at 1,500$g$, 10 min. Neutrophil lysates were prepared as previously described ([66]). For Western blotting, whole-cell lysates and supernatants were heated for 5 min at 95°C with 4X Laemmli buffer (Bio-Rad) and

the samples loaded on a 12% SDS–PAGE before transfer onto a nitrocellulose membrane using Trans-Blot Turbo System (Bio-Rad). After saturation in 5% non-fat milk/TBS/0.1% Tween, membranes were incubated overnight at 4°C with the primary antibodies listed in Table S2. After washing, the membranes were incubated for 1 h at room temperature with secondary antibodies. Bands were visualized using Clarity Max ECL (Bio-Rad) on a Fusion FX imaging system (Vilber Lourmat). Total proteins were measured after stripping for 12 min with Restore WB Stripping Buffer (Thermo Fisher Scientific), using GAPDH (D16H11) XP Rabbit mAb (Cell Signaling Technology).

### Statistical analysis

Individual data and medians are presented in the figures. Statistical analyses were performed using Prism 6.0 software (GraphPad). Analyses were performed on data from two to six independent experiments. Mann–Whitney non-parametric tests or two-way ANOVA tests were used. For Fig 6H, principal component analysis was performed using RStudio with the factoMineR package. For Fig 7H, the Rcmd plugin was used to analyze data as a stratified test. A paired, non-parametric, two-tailed, K-sample Fisher–Pitman permutation test was used to analyze data, with a Monte Carlo resampling approximation. Represented *P*-values are as follows: $*P < 0.05$, $**P < 0.01$, and $***P < 0.001$.

## Data Availability

All data are available in the main text or supplementary materials. Transcriptomic data are available using the BioProject accession number PRJNA1026083.

## Supplementary Information

## Acknowledgements

The mouse team of the PFIE (INRAE, Nouzilly), especially Corinne Beaugé, Jérôme Pottier, Emilie Lortscher, and Laetitia Mérat, is warmly acknowledged for attentive mouse care and prompt reply to all our requests for mouse experiments. We thank Valérie Quesniaux (INEM, UMR7355 CNRS, Université d'Orléans, France) who generously provided bone marrow from *Aim2*$^{-/-}$, *Gsdmd*$^{-/-}$, *Nlrp3*$^{-/-}$, *Csp1/11*$^{-/-}$ mice, and for helpful discussions. Alix Sausset and Christelle Rossignol, from the IMI team (ISP, INRAE, Nouzilly) are acknowledged for their assistance respectively with neutrophil cell sorting and histology. Warm thanks to Sonia Lacroix-Lamandé and the AIM team (ISP, INRAE, Nouzilly) for primers for the Fluidigm BioMark. We thank Roland Brosch (Institut Pasteur, Paris) who kindly provided the HN878 Mtb strain, and Sébastien Leclercq (ISP, INRAE, Nouzilly, France) who verified its genome sequence. Anti-PD-L1 Ab (Tecentriq, atezolizumab) was graciously given by André Rieutord and Hail Aboudagga (Gustave Roussy Cancer Campus, Villejuif, France). Finally, we deeply thank Mustapha Si-Tahar and Hervé Watier (CEPR, UMR 1100, INSERM, Université de Tours, France) for helpful discussions. This work was supported by a FEDER/Region Centre Val de Loire

EuroFéRI grant (FEDER-FSE Centre Val de Loire 2014-2020, N° EX 010233) and IMAG'ISP (FEDER-FSE Centre Val de Loire 2019-2023, N° EX 004654). Travel between France and Brazil for JS Fahel, E Doz-Deblauwe, and N Winter and from Brazil to France for SC Oliveira was supported by the Program Hubert Curien CAPES-COFECUB.

## Author Contributions

E Doz-Deblauwe: conceptualization, formal analysis, supervision, investigation, visualization, methodology, project administration, and writing—original draft, review, and editing.
B Bounab: conceptualization, formal analysis, validation, investigation, visualization, methodology, and writing—original draft, review, and editing.
F Carreras: validation, investigation, and visualization.
JS Fahel: validation, investigation, and visualization.
SC Oliveira: resources, supervision, and writing—review and editing.
M Lamkanfi: resources and writing—original draft.
Y Le Vern: formal analysis and visualization.
P Germon: formal analysis and writing—review and editing.
J Pichon: formal analysis, investigation, and methodology.
F Kempf: software, formal analysis, and methodology.
C Paget: conceptualization and writing—review and editing.
A Remot: conceptualization, formal analysis, visualization, project administration, and writing—original draft, review, and editing.
N Winter: conceptualization, supervision, funding acquisition, visualization, project administration, and writing—original draft, review, and editing.

## Conflict of Interest Statement

The authors declare that they have no conflict of interest.

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
