## [Reviewer comments · Life Science Alliance]

Life Science Alliance

Dual neutrophil subsets exacerbate or suppress inflammation in tuberculosis via-IL-1b or PD-L1

Emilie Doz-Deblauwe, Badreddine Bounab, Florence Carreras, Julia Silveira Fabel, Sergio Oliveira, Mohamed Lamkanfi, Yves Le Vern, Pierre Germon, Julien Pichon, Florent Kempf, Christophe Paget, Aude Remot, and Nathalie Winter

DOI: <https://doi.org/10.26508/lsa.202402623>

Corresponding author(s): *Nathalie Winter, INRAE*

Review Timeline:

Submission Date:	2024-01-26
Editorial Decision:	2024-03-01
Revision Received:	2024-03-27
Editorial Decision:	2024-04-16
Revision Received:	2024-04-26
Accepted:	2024-04-26

Scientific Editor: *Eric Sawey, PhD*

Transaction Report:

March 1, 2024

Re: Life Science Alliance manuscript #LSA-2024-02623-T

Dr. Nathalie Winter
INRAE
INRAE Centre Val de Loire unité ISP
NOUZILLY 37380
France

Dear Dr. Winter,

Thank you for submitting your manuscript entitled "Neutrophil subsets play dual roles in tuberculosis by producing inflammasome dependent-IL-1b or suppressing T-cells via PD-L1" to Life Science Alliance. The manuscript was assessed by expert reviewers, whose comments are appended to this letter. We invite you to submit a revised manuscript addressing the Reviewer comments.

Thank you for this interesting contribution to Life Science Alliance. We are looking forward to receiving your revised manuscript.

Sincerely,

B. MANUSCRIPT ORGANIZATION AND FORMATTING:

Reviewer #1 (Comments to the Authors (Required)):

The investigation adds a layer of complexity to the roles played by neutrophils in TB and may explain the reactivation of this disease observed in cancer patients treated with anti-PD-L1. The findings are extremely interesting, and the paper is well-written. I have the following major comments to improve the manuscript.

Major Comments:

1. The novelty of the work needs little more description in the Introduction and Discussion sections. The limitations of mouse TB with regard to granuloma formation can be mentioned.
2. Were data tested for normality? If so, this should be included in the methods, including which test was used. If not, it is not clear whether the student's t-test is the correct statistical test to use as a non-parametric test like Mann-Whitney would be more appropriate if data did not have a normal distribution.

Reviewer #2 (Comments to the Authors (Required)):

The authors investigated roles and heterogeneity of neutrophils in tuberculosis (TB). Heterogeneity of neutrophils is a timely topic and has not been comprehensively addressed in TB. Combining infection studies, ex vivo analyses, knock-out mouse models and cell transfer experiments the authors conclude that virulent mycobacteria mobilize at least two distinct neutrophil subsets during acute infection of the lung. These neutrophil subsets encompass: 1) pro-inflammatory, IL-1b producing cells (Ly6G+MHCII-); and 2) regulatory, i.e. T cell suppressing (Ly6G+MHCII+PD-L1hi) cells, which "accelerate" or act as a "brake" on the TB pathology, respectively. The dynamics of these two neutrophil populations was also investigated during infection with distinct Mtb lineages and in TB-susceptible IFNg KO mice. Bacterial virulence distinctly modulated frequencies of these subsets, as did absence of IFNgR. Overall, these observations are very interesting and relevant for TB immunology. However, the data do not fully support the conclusions, please see specific concerns below. The title should be amended as there is no data showing that the regulatory subset suppresses T cells in TB.

Figure 1: The authors indicated a 95% purity of the bone marrow neutrophil preparation. To what extent contaminants may actually contribute to the IL-1b release, e.g. monocytes? This is particularly relevant for the BCG observations as these diverge from the in vivo BCG findings.

It is misleading to present BCG and Mtb data head-to-head if various MOI were used.

The requirements for bacterial virulence factors, i.e. RD1 encoded, for the release of mature IL-1b release by macrophages were repeatedly demonstrated, e.g. PMID: 21784976. It is surprising that BMDM produced abundant IL-1b following BCG infection.

Figure 2: Efficiency of the neutrophil depletion should be validated using markers not used for depletion. The lower IL-1b concentration in the MRP8Cre+ mice may stem from distinct frequencies in myeloid cells, aka IL-1b producers beyond neutrophils, as shown in figure 5.

Data derived from one experiment questions reproducibility of the data.

Figure 3: Details on quality of the scRNA-seq data, e.g. viability before processing, is missing.

Do regulatory neutrophils suppress Mtb-specific T cells? Or T cells in TB? Where do these suppressive neutrophils locate within the tissue, i.e. in the proximity of T cells?

Figure 4: It is cumbersome to conclude on the findings as CFU differ significantly at DPI 21 between mice infected with H37Rv vs. HN878. As such, variability in the neutrophil subsets may be due to strain-specific differences or to variable bacterial loads. PD-L1 MFI and frequencies of PD-L1 expressing MHC+ and MHC- subsets should be presented.

Figure 5: The title states that IL-1b produced by inflammatory neutrophils sustains inflammation. Lung inflammation seems not to differ in MPRCR+ mice, although quantification should be provided for panel I. Again, lower IL-1b may be due to lower lung frequencies of cells able to produce IL-1b (due to recruitment or cell death).

It is unclear what cells are shown in panel H, alveolar macrophages, monocyte-derived macrophages or - dendritic cells, interstitial lung macrophages? Based on CD11b and CD11c it is difficult to accurately define such cell subsets. What are MPs?

Figure 6: As in figure 4, variability in CFU between the mouse strains interferes with the other readouts. Are regulatory, MHCII+ neutrophils, significantly more in IFNg KO mice? Do the subsets differ in their viability features? IFNgR has been reported to regulate neutrophil viability in TB, PMID: 21967766.

Figure 7: The viability of the cell post transfer should be considered (see also above). Differences in viability, not only cell recruitment, should be considered for the PD-L1 expressing subsets. Does IFNg, or STAT signaling, alter PD-L1 expression on neutrophils?

The purity of neutrophils isolated based on untouched magnetic sorting should be indicated (line 630), as well as their downstream usage.

Typos (Ilr2 instead of Il1r2 multiple times, .e.g. line 209) and language (e.g. pathophysiology; not sure what is meant by "conjugated paths", line 495) should be carefully checked. Phrasing should be carefully checked to avoid any overstatement, e.g. lines 488-489 (no data on severe forms of TB, but mouse data), lines 509-510 (pathology was not significantly different).

Reviewer #1 (Comments to the Authors (Required)):

The investigation adds a layer of complexity to the roles played by neutrophils in TB and may explain the reactivation of this disease observed in cancer patients treated with anti-PD-L1. The findings are extremely interesting, and the paper is well-written. I have the following major comments to improve the manuscript.

We thank our reviewer for his kind words and interest in our work.

Major Comments:

1. The novelty of the work needs little more description in the Introduction and Discussion sections. The limitations of mouse TB with regard to granuloma formation can be mentioned.

We modified the introduction (lines 81-84) and the discussion (lines 348-355) to better underline the novelty of our findings. We have also emphasized the limits of the classical mouse models regarding TB granuloma formation (discussion, lines 432-436).

2. Were data tested for normality? If so, this should be included in the methods, including which test was used. If not, it is not clear whether the student's t-test is the correct statistical test to use as a non-parametric test like Mann-Whitney would be more appropriate if data did not have a normal distribution.

Since we mostly used *in vivo* experimentation in mice in this work, which is subjected to high regulation by ethics committees, all the data obtained were from experiments with small numbers of individuals. By definition normal distribution of data does not apply in that case. Thus, we only used non-parametric statistical tests that are suited to small samples sizes. The Student's test was not used in our statistical analyses (as indicated in all figures legends). However, we indeed used ANOVA which was inappropriate. We changed statistical analysis and used the appropriate non-parametric Fisher-Pitman permutation test throughout the study which did not modify statistical significance. We deeply apologize for this mistake.

Reviewer #2 (Comments to the Authors (Required)):

The authors investigated roles and heterogeneity of neutrophils in tuberculosis (TB). Heterogeneity of neutrophils is a timely topic and has not been comprehensively addressed in TB. Combining infection studies, ex vivo analyses, knock-out mouse models and cell transfer experiments the authors conclude that virulent mycobacteria mobilize at least two distinct neutrophil subsets during acute infection of the lung. These neutrophil subsets encompass: 1) pro-inflammatory, IL-1b producing cells (Ly6G+MHCII-); and 2) regulatory, i.e. T cells suppressing (Ly6G+MHCII+PD-L1hi) cells, which "accelerate" or act as a "brake" on the TB pathology, respectively. The dynamics of these two neutrophil populations was also investigated during infection with distinct Mtb lineages and in TB-susceptible IFNg KO mice. Bacterial virulence distinctly modulated frequencies of these subsets, as did absence of IFNgR. Overall, these observations are very interesting and relevant for TB immunology. However, the data do not fully support the conclusions, please see specific concerns below. The title should be amended as there is no data showing that the regulatory subset suppresses T cells in TB.

We thank our reviewer for his interest in our work. As requested, we have changed the title which is now "**Dual neutrophil subsets exacerbate or suppress inflammation in tuberculosis via IL-1 β or PD-L1**"

Figure 1: *The authors indicated a 95% purity of the bone marrow neutrophil preparation. To what extent contaminants may actually contribute to the IL-1b release, e.g. monocytes?*

Although we cannot completely rule out minor contamination during preparation of neutrophils from the bone marrow, we do not believe that our observation of IL-1 β production results from other myeloid populations. If this was the case, we would not observe the complete extinction of IL-1 β production by neutrophils prepared from the bone marrow of *MRP8^{Cre+}Csp1^{fllox}* in comparison to *MRP^{WT}Csp1^{fllox}* littermates after stimulation with LPS+Nig (Figure 2F).

We also provide for our reviewer only, the data of one experiment (see on the left) where we measured IL-1 β production by bone-marrow neutrophils or macrophages from *MRP8^{Cre+}Csp1^{fllox}* and *MRP^{WT}Csp1^{fllox}* after stimulation with BCG. Again, complete extinction of IL-1 β production by neutrophils from *MRP8^{Cre+}Csp1^{fllox}* was observed in comparison with *MRP^{WT}Csp1^{fllox}* cells. No difference was observed in macrophages prepared from the same animals. We believe that if there

was contamination in neutrophil preparations with some monocytes or macrophages we would detect IL-1 β , since these cells would have a functional Csp1 and therefore would be able to assemble the inflammasome. All bone-marrow neutrophils were prepared in the same conditions throughout our study. Thus, we do not believe that neutrophils preparations from C57BL/6 mice and different knock-out used in Figure 1 would be contaminated by other IL-1 β producing myeloid-cells while preparations from *MRP8^{Cre+}Csp1^{fllox}* mice would not.

We also remind our reviewer that IL-1 β production by neutrophils was also observed by another method i.e. flow cytometry (Figure 3G) : using intracellular staining with an antibody specific for mature IL-1 β , we clearly detected IL-1 β in MHC-II^{neg} neutrophils and not in MHC-II^{pos} regulatory neutrophils.

This is particularly relevant for the BCG observations as these diverge from the in vivo BCG findings.

Indeed, there are differences regarding IL-1 β production by neutrophils after *in vitro* stimulation by BCG or infection *in vivo*. As we propose in the discussion (lines 369 -371) we rather believe that this is due to “danger” signals that are not sensed the same way by neutrophils *in vitro* and *in vivo*. We explain this “discrepancy” by the fact that the “danger” signal 2 is not strong enough to assemble the inflammasome after BCG infection *in vivo*.

It is misleading to present BCG and Mtb data head-to-head if various MOI were used.

We carefully specified in the figure legends, as well as in text in the results section, that different MOI were used to infect neutrophils with BCG (non-virulent) and Mtb (virulent) to preserve neutrophil viability. Thus, we did not intentionally try to mislead our reviewer and we apologize if he/she feels this is the case. We introduced a break-line on the axis to emphasize the different MOI in the graph (revised Figure 1).

The requirements for bacterial virulence factors, i.e. RD1 encoded, for the release of mature IL-1 β release by macrophages were repeatedly demonstrated, e.g. PMID: 21784976. It is surprising that BMDM produced abundant IL-1 β following BCG infection.

It is indeed established that the RD1 region is an important trigger of inflammation -and IL-1 β production- by macrophages. However, we are not the only ones to report IL-1 β production by macrophages infected with BCG. In reference cited by our reviewer (PMID: 21784976), it is true that IL-1 β production by human macrophages or THP1 was higher after H37Rv than BCG infection (used at MOI of 5 in this work

whereas we used MOI of 10). However, the authors did detect small amounts of IL-1 β after BCG infection (Figure 6A, C and D).

In the following references, IL-1 β production by BCG-infected macrophages was also reported:

- Paloma Rezende Corrêa *et al.*, compared BCG Pasteur and Moreau (MOI 10) and reported respectively 150 and 100ng/mL of IL-1 β in the supernatant at 6h post infection (PMID: 37851722).
- Wei-Wei Tian *et al.*, measured approximately 250ng/mL of IL-1 β after 24h infection of monocyte derived human macrophages with the BCG strain TMC 1010 (PMID: 23930073).
- O'Shea *et al.*, reported 645ng/mL of IL-1 β produced by BMDM infected with BCG, MOI 10, at 72h post-infection (PMID: 26586698).
- We also reported IL-1 β production by bone-marrow derived macrophages in the past and used the same protocols to prepare and infect macrophages in the present work (Doz-Deblauwe *et al.* PMID: 31921172).

Therefore, we are convinced that macrophages are able to produce mature IL-1 β after BCG infection, which probably means that RD1 is not the only trigger for the inflammasome assembly in these cells.

Figure 2: Efficiency of the neutrophil depletion should be validated using markers not used for depletion. The lower IL-1 β concentration in the MRP8 $^{Cre+}$ mice may stem from distinct frequencies in myeloid cells, aka IL-1 β producers beyond neutrophils, as shown in figure 5.

The efficacy of neutrophil depletion *in vivo* using anti-Ly-6G antibody (clone NIMPR14 or 1A8) has been largely validated in the past both by ourselves (PMID 26871571) and many others (for example PMID 27375607 ; 22264515 ; 23853593 ; 2155529). So, we are confident in our protocol and accuracy of our results.

We provide for our reviewer only, extra FACS analysis of the cellular composition in the lung after depletion. The number of CD11b+ cells was clearly decreased after anti-Ly-6G inoculation in mice. Similarly, the number of cells in the CD11b and granulocyte-gate (analyzed by FSC ad SSC) was also highly decreased in depleted mice. We hope these supplementary analyses will convince our reviewer that neutrophils were efficiently depleted.

Data derived from one experiment questions reproducibility of the data.

As stated above the protocol has been largely validated in the past and, in line with ethical committees, we did not want to use too many animals for incremental results. The real novel part in Figure 2 was panel D showing reduction of total IL-1 β in lung tissue in antibody versus isotype treated mice, **suggesting** contribution of neutrophils to IL-1 β production *in vivo* during TB. However, antibody depletion had clear limits to explore further the true contribution of neutrophils to IL-1 β -mediated inflammation *in vivo*. This is why we then concentrated our efforts -and *in vivo* animal experimentation- on the new MRP8 $^{Cre+}$ Csp1 lox mouse model where IL-1 β production was specifically depleted in neutrophils. All experiments with this model were repeated twice, at least.

Figure 3: Details on quality of the scRNA-seq data, e.g. viability before processing, is missing.

We agree with our reviewers that the gating strategy presented in Figure S2A did not clearly state that a viability dye was always used in our experiments to exclude dead cells (neutrophils) from all our *ex vivo* experiments. We now provide a revised version of the gating strategy (see Revised Figure S2A, viability plot has been included). We apologize for this mistake.

Do regulatory neutrophils suppress Mtb-specific T cells?

We did not run such experiments. Although we doubt that regulatory neutrophils would suppress OT2 cells and not TB-specific T cells.

Or T cells in TB?

Yes, this is the case as we showed in figure 7 and S2 B and C. Regulatory neutrophils from IFN- γ R wild type mice (i.e. PD-L1^{hi}) have a strong impact in the number of T cells in the lung, which we interpret as a direct suppressive effect on Mtb-specific T cells *in vivo*.

Where do these suppressive neutrophils locate within the tissue, i.e. in the proximity of T cells?

Actually, we did not yet completely dissect out the mechanism by which regulatory neutrophils interact with T cells. This work is ongoing and we believe this is beyond the scope of the current study.

Figure 4: *It is cumbersome to conclude on the findings as CFU differ significantly at DPI 21 between mice infected with H37Rv vs. HN878. As such, variability in the neutrophil subsets may be due to strain-specific differences or to variable bacterial loads.*

We are sorry that our reviewer finds our findings difficult to interpret. We never claimed that high CFUs numbers were not correlated to exacerbated neutrophil recruitment and inflammation observed with the HN878 strain. Actually, we are convinced that high CFUs numbers are part of the high virulence profile of the HN878 strain (as also shown by many others). The interesting finding in this part of the work is the down modulation of PD-L1 expression on regulatory neutrophils by this hypervirulent strain. Indeed, we were rather surprised to observe higher recruitment of MHC-II^{pos} neutrophils in the lungs by this strain as compared to the less virulent strain while conventional (i.e. inflammatory) neutrophils remained comparable (Figure 4G). This was rather counterintuitive since lung inflammation was exacerbated with HN878. However, we had an explanation when we discovered that the level of expression of PD-L1 on regulatory neutrophils was dramatically reduced in HN878 versus H37Rv infected mice. Of course, we do not know yet what are mechanisms behind these findings. They will be explored in another study.

PD-L1 MFI and frequencies of PD-L1 expressing MHC+ and MHC- subsets should be presented.

These data have been added (Revised Figure 4I). There was no difference in frequency of regulatory neutrophils between the two different Mtb strains, we only observed a strong difference in PD-L1 surface expression (MFI).

Figure 5: *The title states that IL-1b produced by inflammatory neutrophils sustains inflammation. Lung inflammation seems not to differ in MPRCR+ mice, although quantification should be provided for panel I.*

The quantification of lesioned lung surface in *MRP8^{Cre+}Csp1^{flox}* and *MRP^{WT}Csp1^{flox}* mice at day 21 has been added (Revised Fig. 5J). Indeed, there is no statistically significant difference in the surface occupied by lesions in the lung at that time point between the two mouse lines. However, we would like to emphasize that wild type and genetically modified C57BL/6 mice models, that are most widely used in TB today, have clear limits in terms of physiopathology studies as they do not form proper granulomas (as also stated by Reviewer 1 who requested some changes in the text along those lines). Thus, it is difficult to observe discrete impact of the overall complex immunological response on lesions

development in such models. Moreover, we had to limit our study to one time point (Day 21) due to limited budget (BSL3 animal facilities and labs have very high costs). Later time points would certainly help to observe impact of IL-1 β produced by neutrophils *in vivo* on lung lesions evolution. However, it is commonly admitted in these models that the early cellular infiltrate is a relevant signature of “inflammation” induced by Mtb infection. Using this proxy, we clearly observed “dampening of inflammation” when neutrophils were unable to produce IL-1 β *in vivo* after Mtb infection. Indeed, total CD45^{pos} leukocytes (Figure 5D), neutrophils (Figure 5E) and the two subsets (Figure 5F) as well as CD4 T-cells (Figure 5G) were all diminished in *MRP8^{Cre+}Csp1^{flox}* as compared to *MRP^{WT}Csp1^{flox}* littermates *MRP8^{Cre+}Csp1^{flox}*. So, we believe that we did not misinterpret our data and that IL-1 β produced by conventional inflammatory neutrophils sustains lung inflammation during TB.

Again, lower IL-1b may be due to lower lung frequencies of cells able to produce IL-1b (due to recruitment or cell death).

We never claimed the opposite. This is indeed a possibility. Our only claim is that IL-1 β produced by conventional inflammatory neutrophils sustains lung inflammation during TB.

It is unclear what cells are shown in panel H, alveolar macrophages, monocyte-derived macrophages or - dendritic cells, interstitial lung macrophages? Based on CD11b and CD11c it is difficult to accurately define such cell subsets. What are MPs?

Actually, we referred here to all CD11b^{pos} cells that were Ly-6G^{neg} (i.e. not neutrophils) This is, to the best of our knowledge, mainly composed of macrophages (MPs), but we agree with our reviewer that the definition of these cells was not sharp. In Revised Figure 5H we changed the axis legend “MPs” for “other CD11b^{pos} cells”.

Figure 6: *As in figure 4, variability in CFU between the mouse strains interferes with the other readouts.*

As explained above, we do not claim that the differences observed regarding inflammation are not connected to CFUs. What we would like to emphasize here is more the imbalance between the two neutrophil subsets in IFN- γ R^{-/-} animals as compared to wild type plus the fact that regulatory neutrophils from IFN- γ R^{-/-} mice keep MHC-II but loose PD-L1 expression.

Are regulatory, MHCII+ neutrophils, significantly more in IFNg KO mice?

As shown in Figure 6D, while recruitment of conventional inflammatory neutrophils was highly and significantly increased in IFN- γ R^{-/-} mice as compared to wild type, there was only a trend toward more elevated numbers of MHC-II+ neutrophils and the difference did not reach statistical significance. Moreover, these MHC-II+ neutrophils lost PD-L1 expression (as shown in Figure 7 B and C).

Do the subsets differ in their viability features? IFN-gR has been reported to regulated neutrophil viability in TB, PMID: 21967766.

We thank our reviewer for this important point. We have added viability data in revised Supplementary Figures S4 and S5. Indeed, the percentage of live neutrophils was more important in IFN- γ R^{-/-} mice than in wild type as already observed by Nandi and Behar (PMID: 21967766). Interestingly this difference in survival was observed both for MHC-II^{pos} regulatory and MHC-II^{neg} conventional neutrophils. The percentage of viable regulatory and conventional neutrophils or CD3 T-cells recovered from the lungs of IFN- γ R^{-/-} Mtb infected mice that were mock treated or that received regulatory neutrophils from wild type mice was not different though. Therefore, this reinforces our point: despite higher viability of both subsets of neutrophils in IFN- γ R^{-/-} mice, we observed diminished recruitment of CD3+ T cells (Fig. 7K)

and conventional neutrophils (Fig. 7L) only in the group transferred with regulatory neutrophils and not in the mock-treated group.

Figure 7: *The viability of the cell post transfer should be considered (see also above).*

Viability dye was included in all our sort experiments (as shown in gating strategy from revised Figure S2). Only viable regulatory neutrophils cells were purified from BCG-infected wild type mice and transferred into IFN- γ R^{-/-} Mtb infected recipient mice.

Differences in viability, not only cell recruitment, should be considered for the PD-L1 expressing subsets.

See also our response above. We only transferred viable regulatory neutrophils into recipient IFN- γ R^{-/-} infected recipient mice. Although we cannot rule out that once transferred these cells quickly died, they still enough time to exert inflammation-dampening effects that we reported in Figure 7 G to O). We did not observe any difference in T-cell viability among the groups.

Does IFN γ , or STAT signaling, alter PD-L1 expression on neutrophils?

We did not yet examine the mechanisms involved in T-cell suppression and inflammation release exerted by regulatory neutrophils. Although the suggestions made by our reviewer are worth considering we believe this is beyond the scope of the present study.

The purity of neutrophils isolated based on untouched magnetic sorting should be indicated (line 630), as well as their downstream usage.

Thanks for this reminder; the mistake has been corrected

Typos (Ilr2 instead of Il1r2 multiple times, .e.g. line 209) and language (e.g. pathophysiology; not sure what is meant by "conjugated paths", line 495) should be carefully checked. Phrasing should be carefully checked to avoid any overstatement, e.g. lines 488-489 (no data on severe forms of TB, but mouse data), lines 509-510 (pathology was not significantly different).

Thanks for these remarks. Typos, language and phrasing have been corrected accordingly.

April 16, 2024

RE: Life Science Alliance Manuscript #LSA-2024-02623-TR

Dr. Nathalie Winter
INRAE
INRAE Centre Val de Loire unité ISP
NOUZILLY 37380
France

Dear Dr. Winter,

Thank you for submitting your revised manuscript entitled "Dual neutrophil subsets exacerbate or suppress inflammation in tuberculosis via-IL-1b or PD-L1". We would be happy to publish your paper in Life Science Alliance pending final revisions necessary to meet our formatting guidelines.

- please be sure that the authorship listing and order is correct
- please upload your Tables in editable .doc or excel format
- please move the references section before the figure legends
- please be sure that all authors are mentioned in the Authors Contribution section in the manuscript text
- please add a callout for Figure 4F to your main manuscript text

FIGURE CHECKS:

- please add a scale bar to Figure S2B

A. FINAL FILES:

B. MANUSCRIPT ORGANIZATION AND FORMATTING:

Sincerely,

Reviewer #2 (Comments to the Authors (Required)):

The authors satisfactorily addressed most review comments by revising the manuscript and adding new data to cover relevant questions.

April 26, 2024

RE: Life Science Alliance Manuscript #LSA-2024-02623-TRR

Dr. Nathalie Winter
INRAE
INRAE Centre Val de Loire unité ISP
NOUZILLY 37380
France

Dear Dr. Winter,

Thank you for submitting your Research Article entitled "Dual neutrophil subsets exacerbate or suppress inflammation in tuberculosis via-IL-1b or PD-L1". It is a pleasure to let you know that your manuscript is now accepted for publication in Life Science Alliance. Congratulations on this interesting work.

DISTRIBUTION OF MATERIALS:

Again, congratulations on a very nice paper. I hope you found the review process to be constructive and are pleased with how the manuscript was handled editorially. We look forward to future exciting submissions from your lab.

Sincerely,
